# CALM 🪶 Before the STORM ⚡:
# Unlocking Native Reasoning for Optimization Modeling

## Abstract

Large Reasoning Models (LRMs) have demonstrated strong capabilities in complex multi-step reasoning, opening new opportunities for automating optimization modeling. However, existing domain adaptation methods, originally designed for earlier instruction-tuned models, often fail to exploit the advanced reasoning patterns of modern LRMs — In particular, we show that direct fine-tuning on traditional *non-reflective* datasets leads to limited gains. To fully leverage LRMs' inherent reasoning abilities, we propose **CALM** (*Corrective Adaptation with Lightweight Modification*), a framework that progressively refines LRMs within their native reasoning modes for optimization modeling tasks. In CALM, an expert intervener identifies reasoning flaws and provides concise corrective hints, which the LRM incorporates to produce improved reasoning trajectories. These interventions modify fewer than 2.6% of generated tokens, but generate high-quality data for soft adaptation through supervised fine-tuning. The adapted model is then further improved through reinforcement learning. Building on CALM, we develop **STORM** (*Smart Thinking Optimization Reasoning Model*), a 4B-parameter LRM that achieves a new state-of-the-art average accuracy of 68.9% across five popular optimization modeling benchmarks, matching the performance of a 671B LRM. These results demonstrate that dynamic, hint-based data synthesis both preserves and amplifies the native reasoning patterns of modern LRMs, offering a more effective and scalable path towards expert-level performance on challenging optimization modeling tasks.

## 1 Introduction

Operations Research (OR) and optimization modeling techniques are central to decision-making in areas such as inventory management and airline crew scheduling (Silver, 1981; Vance et al., 1997). Yet, despite their importance, the translation of real-world problems into mathematical models has long been a bottleneck, as it requires substantial human expertise (Huang et al., 2025). In this context, Large Language Models (LLMs) introduce a promising path toward automation. With the advent of instruction-tuned models, early works such as ORLM (Huang et al., 2025), LLMOPT (Jiang et al., 2024), and Solver-Informed RL (Chen et al., 2025) made notable progress. These methods establish a prevailing paradigm: constructing *non-reflective datasets* and training LLMs for direct generation of an optimization model and its solver code from a problem description (see Figure 1a for an example). Here, we refer to a *non-reflective dataset* as a pre-collected set of static problem–solution pairs without intermediate reasoning or feedback.

However, the emergence of Large Reasoning Models (LRMs) represents a new paradigm in the field. Unlike LLMs, LRMs possess an inherent capacity for multi-turn reasoning, which we call their *native reasoning patterns*. This capability allows iterative and adaptive reasoning within a single inference pass (Qwen Team, 2025; DeepSeek-AI, 2025), offering greater flexibility than traditional non-reflective generation.

Although existing methods can still be applied to LRMs (Huang et al., 2025; Jiang et al., 2024), it exhibits some misalignments. On the one hand, they neglect the native reasoning patterns of these models, imposing artificial reasoning modes instead. On the other hand, their data synthesis strategies remain non-reflective, which conflicts with the dynamic reasoning loops that characterize LRMs. As we empirically demonstrate in Section 2, these misalignments may provide only marginal improvements and fail to fully exploit the potential of LRMs.

These observations naturally lead to a central research question: *Can we leverage the native reasoning of LRMs to solve optimization modeling tasks effectively?* Answering this question is essential for advancing the application of LRMs, especially as high-performance open-source variants become increasingly available.

To address this question, we design an evaluation protocol to systematically examine the flaws in native reasoning patterns for optimization modeling tasks. The evaluation reveals seven recurring flaws types, which we categorize into two groups: (1) *Code Utilization Distrust* and (2) *Lack of OR Expertise*. While the latter has been discussed in prior work (Huang et al., 2025; Jiang et al., 2024), the former remains largely overlooked in research on automated optimization modeling.

These flaws provide a natural entry point for method design. In response, we introduce **CALM** (*Corrective Adaptation with Lightweight Modification*), a framework that uses lightweight intervention to adapt LRM reasoning trajectories, aligning their native reasoning patterns with the requirements of optimization modeling tasks. Two features make this framework particularly effective. First, inspired by Li et al. (2025a), we allow the LRM to access a solver's code compiler, providing immediate execution feedback and thereby strengthening reflective reasoning — an ability absent in typical LRMs and earlier approaches. Second, the interventions are deliberately lightweight, accounting for fewer than 2.6% of the total tokens.

The expert-level trajectories generated by CALM support a two-stage training pipeline: supervised fine-tuning for soft adaptation of reasoning habits, followed by reinforcement learning to refine these skills and achieve autonomous mastery. The final model is denoted as **STORM** (*Smart Thinking Optimization Reasoning Model*).

Our contributions are as follows:

- We provide empirical evidence on the limitations of adapting modern LRMs via fine-tuning on non-reflective datasets, highlighting the importance of preserving their native reasoning patterns.

- We propose **CALM**, a lightweight and scalable framework that leverages solver code execution to correct and strengthen LRM reasoning trajectories, aligning it with the demands of optimization modeling tasks.

- Our final model, **STORM**, with 4B parameters, sets a new state of the art across five optimization modeling benchmarks, matching the performance of a 671B LRM.

- Our controlled analysis of reinforcement learning reveals that CALM-based adaptation is crucial for success. The adapted model learns faster and reaches a higher performance ceiling, driven by a shift to a computation-driven reasoning pattern that enables it to more effectively build and refine expert-level optimization modeling skills.

We situate our work within the broader literature and provide a discussion of related work in Appendix A. We also clarify the role of LLMs in the preparation of this manuscript in Appendix K.

## 2 BACKGROUND AND MOTIVATION

### 2.1 BACKGROUND: LLMs FOR OPTIMIZATION MODELING

Automated optimization modeling is the task of translating a natural language problem description into a mathematical model and executable solver code (see Figure 1a). For evaluation, the solver computes a candidate solution, which is deemed correct if its objective value lies within a predefined relative error of the ground truth. Performance is assessed on benchmarks that span a range of difficulty, from easy problems in NL4Opt to complex industrial cases in IndustryOR. A detailed overview of these benchmarks is provided in Appendix E.1. As shown in Figure 1b, this task can be approached through two mainstream paradigms that differ fundamentally in how the final solution is obtained.

**Non-reflective Generation.** Early methods, particularly those based on traditional LLMs, approach optimization modeling as a *non-reflective generation* problem (Huang et al., 2025; Jiang et al., 2024). As shown in Figure 1b (top), the LLM receives a problem description and generates a complete solution in a single step, including both the mathematical model and the solver code. The reasoning process is linear, with no opportunity for feedback or revision based on solver execution results.

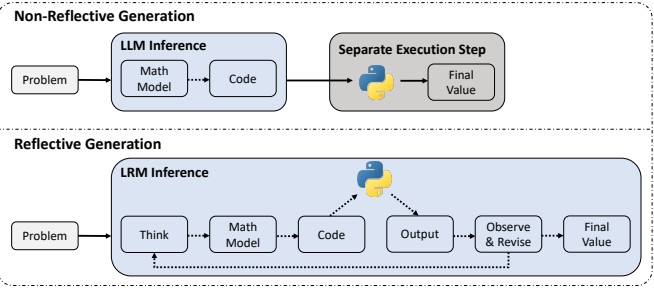

**(a)** An optimization modeling example. Full details in Appendix B.

**(b)** Comparison of reasoning paradigms in automated optimization modeling.

Figure 1: Illustrations of optimization modeling and reasoning paradigms.

**Reflective Generation.** The advent of modern LRMs (Qwen Team, 2025; DeepSeek-AI, 2025) has introduced a new paradigm. These models exhibit a range of sophisticated reasoning patterns, with *reflective generation* — the capacity for iterative self-correction and refinement — emerging as a dominant mode (Jaech et al., 2024). We thus treat this as the primary reasoning pattern for LRMs in our study, as it is well-suited for optimization modeling, which often requires numerical feedback and mirrors the trial-and-error process of human experts. Accordingly, we design a reasoning workflow that integrates solver feedback into this reflective process, as shown in Figure 1b (bottom). In this paradigm, LRMs behave more like human experts operating in an interactive environment. They can propose hypotheses, generate code, execute it, observe outputs, and refine their reasoning accordingly.

## 2.2 PILOT STUDY: ADAPTING LRMS WITH NON-REFLECTIVE DATA

Given the availability of open-source LRMs and well-established non-reflective datasets from prior work (Huang et al., 2025; Lu et al., 2025), a natural first step is to test the most direct adaptation strategy: fine-tuning an LRM on these existing datasets. This pilot study provides a necessary baseline and examines whether such training improves performance across tasks of varying difficulty.

Table 1: Performance of a base LRM before and after SFT on the existing dataset.

| Model | NL4OPT | MAMO Easy | MAMO Complex | IndustryOR | OptMath | Macro AVG |
|---|---|---|---|---|---|---|
| Base LRM | 85.8 | 73.8 | 46.5 | 46.2 | 33.1 | 57.1 |
| + SFT on Non-reflective Data | 92.9 | 88.7 | 40.5 | 27.5 | 6.6 | 51.2 |
| Absolute Change | +7.1 | +14.9 | -6.0 | -18.7 | -26.5 | -5.9 |

The results of our pilot study in Table 1 show a clear trade-off. The LRM achieves higher accuracy on easier tasks such as `MAMO-Easy`, but its performance declines sharply on more complex benchmarks like `IndustryOR` and `OptMath`. The full experimental setup is described in Appendix E.2.

A plausible explanation is that existing datasets contain only problem–solution pairs, which push the LRM to replace its native multi-step reasoning with a rigid, non-reflective generation style it is not optimized for. This shift improves simple cases but undermines the model's reasoning ability on complex tasks, a pattern also reported in other domains (Zhang et al., 2025). This observation highlights the central motivation of our work: **To unlock an LRM's full potential, adaptation must preserve its native reasoning patterns.**

## 2.3 A TAXONOMY OF FLAWS IN LRM'S NATIVE REASONING

Our pilot study confirms that preserving the LRM's native reasoning is essential. This finding, however, raises a further question: are these native patterns sufficient for expert-level performance or require targeted enhancement? To address this, we first need to systematically examine the inherent weaknesses of an unguided LRM in optimization modeling tasks.

**Establishing a Protocol for Flaw Identification.** To perform a rigorous analysis, we establish a systematic protocol. We first prompted a base LRM to generate solutions for a diverse set of problems. A team of human experts with backgrounds in OR then analyzed these responses to identify recurring error patterns. Through a collaborative, multi-stage process of annotation, clustering, and refinement, the team converged on a set of seven distinct flaw types, which form the basis of our taxonomy. The complete, detailed protocol for this human-in-the-loop analysis is provided in Appendix C.

**A Two-Category Taxonomy of Flaws.** Our analysis of the 7 identified flaw types reveals that 6 are **major reasoning flaws**, representing fundamental challenges in the modeling process. The seventh, a minor procedural error, is detailed in Appendix D. Our taxonomy focuses on the 6 major flaws, which we group into two high-level conceptual categories:

- **Code Utilization Distrust:** This category encompasses flaws where the LRM fails to properly leverage the computational solver, such as attempting manual calculations or writing fragmented code (Triggers 1-3). This indicates an inefficient reasoning strategy and an under-reliance on powerful external tools.
- **Lack of OR Expertise:** This category covers fundamental errors in modeling and logic, including flawed mathematical formulations, missed constraints, and implementation errors (Triggers 4-6). These flaws stem from insufficient domain-specific knowledge.

This two-level taxonomy provides a structured framework for understanding and addressing LRM failures.

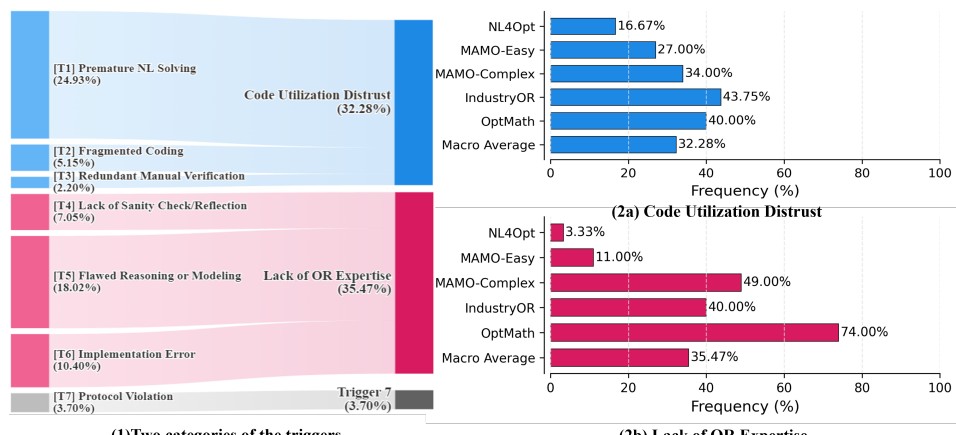

**Figure 2:** Trigger Categorization and Distribution. The left (1) shows the macro-average frequency of each trigger, the first 6 triggers grouped into two primary categories. The right (2a and 2b) detail the frequency distribution of these two main categories across the evaluated benchmarks.

**Quantifying Flaw Distribution across Benchmarks.** With this taxonomy in place, we quantify the prevalence of these flaws at scale using an expert-level LLM as a consistent, automated annotator. Details on this quantification process can be found in Appendix J. The distribution, shown in Figure 2 (2a) and (2b), reveals a critical insight: the primary bottleneck varies with problem difficulty. On easy-to-medium tasks like `NL4Opt` and `MAMO-Easy`, flaws are dominated by *Code Utilization Distrust*. In contrast, on complex benchmarks like `OptMath`, a *Lack of OR Expertise* becomes the main barrier. This reveals the core challenge for effective adaptation: **LRM reasoning must be enhanced to overcome the bottlenecks of inefficient code use and a lack of OR expertise.**

## 3 METHODOLOGY

### 3.1 PRELIMINARIES: FORMALIZING THE REFLECTIVE GENERATION FLOW

We formalize the LRM's problem-solving process as a sequential interaction within a code interpreter environment $E$. Given a problem $P$, the LRM—referred to as the *Reasoner*—generates an iterative Reasoning Flow, represented as

$$\tau^{(T)} = (s_0, a_0, o_0, s_1, a_1, o_1 \ldots, s_T, a_T, o_T), \tag{1}$$

where $s_t, a_t$ are the textual reasoning and code block at step $t$, respectively. The sequential reasoning flow follows these steps:

$$(s_t, a_t) = \pi_\theta(\tau^{(t-1)}), o_t = E(a_t),$$
$$\tau^{(t)} = \tau^{(t-1)} \oplus s_t \oplus a_t \oplus o_t. \tag{2}$$

The objective is to refine $\pi_\theta$ to produce trajectories that ultimately yield correct solutions.

## 3.2 CALM: Correcting Adaptation with Lightweight Modification

At the heart of our approach is the **CALM** framework, a dynamic data curation method based on a *Reasoner–Intervener* collaboration pattern for generating expert-aligned reasoning flows.

**Targeted Hints for Specific Flaws.** CALM's strength lies in its one-to-one mapping between reasoning flaws and tailored hints injected by the Intervener (see Appendix D). These interventions address two primary issues:

- *For Code Utilization Distrust*: When the Reasoner attempts manual solving, the Intervener injects a hint to redirect it toward using the solver, such as: *"Wait, maybe I can use the 'pulp' library and let the solver find the optimal solution."*

- *For Lack of OR Expertise*: When key concepts like integer constraints are missed, the Intervener provides concise domain-specific guidance, such as: *"A fractional number of cars isn't practical, suggesting a missed integer constraint."*

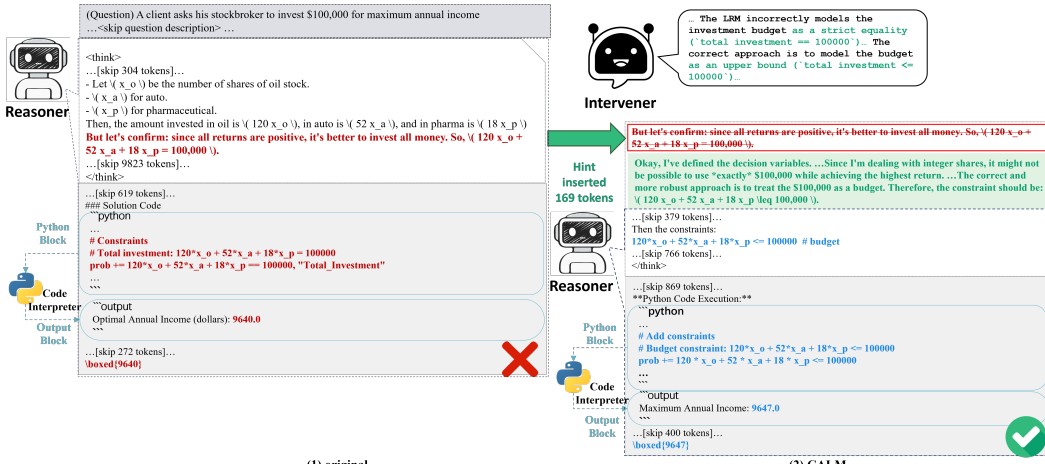

**Figure 3:** A representative example of *Lack of OR Expertise* flaw. (1) The model's native reasoning results in an incorrect problem formulation, leading to a wrong answer. (2) In contrast, the process under **CLAM**'s guidance correct the formulation, enabling the model to find the correct solution.

**The Iterative Hinting Loop.** CALM implements an iterative refinement loop that transforms flawed reasoning trajectories into expert-aligned ones. Let $\tau^{(i)}$ denote the reasoning flow at iteration $i$. The process proceeds as follows:

- **Initial Generation** ($i = 0$)**:** The Reasoner generates an initial trajectory $\tau^{(0)}$ for a given problem $P$.

- **Intervention & Evaluation:** The Intervener examines $\tau^{(i)}$. If no deviation is found, the process terminates with the final trajectory $\tau^* = \tau^{(i)}$. Otherwise, the Intervener identifies the flawed step $t$ and corresponding action $a_t$, and generates a corrective hint $h_i$.

- **Localized Revision & Resumption:** A modified state is formed by appending the hint $h_i$ to the context at step $t$. From this new state, the Reasoner continues its reasoning process to form a corrected trajectory $\tau^{(i+1)}$.

This loop continues until the Intervener deems the reasoning trajectory to be complete and free of flaws. As a practical safeguard to prevent unproductive or infinite correction cycles, we limit the maximum number of interventions. Each intervention is localized and minimally invasive, preserving the model's native reasoning (see Appendix H for more examples of specific, single-step interventions, and Appendix I for a complete, multi-turn case study).

**Filtering of Expert Trajectories.** To ensure supervision quality, we construct our SFT dataset, $\mathcal{D}_{CALM}$, by filtering for "golden" trajectories (Figure 5). We retain only those that are both correct in their final answer and assessed as having a flawless reasoning flow by the Intervener.

### 3.3 Training Pipeline: From Soft Adaptation to Autonomous Mastery

The trajectories curated and filtered by CALM are used in a two-stage training pipeline.

**Stage 1: Supervised Fine-Tuning for Soft Adaptation.** We fine-tune the base LRM on $\mathcal{D}_{CALM}$, using a standard cross-entropy loss. The goal of this stage is not to enhance the final performance score, but to perform a soft adaptation of the policy $\pi_\theta$. By training on trajectories that align with the model's native reasoning, this approach guides its problem-solving habits without constraining it into a rigid, non-reflective pattern.

**Stage 2: Reinforcement Learning for Autonomous Mastery.** Following supervised fine-tuning, we apply reinforcement learning to enable the model to independently optimize for correctness. We use the Group Relative Policy Optimization (GRPO) algorithm (Shao et al., 2024), allowing interaction with the Code Interpreter for up to $T = 4$ code executions per rollout. The RL stage aims to maximize the expected reward: $J(\theta) = \mathbb{E}_{\tau \sim \pi_\theta(\cdot|P)}[R(\tau)]$. Our reward function is a simple binary signal based on the final outcome:

$$R(\tau) = \begin{cases} 1 & \text{if } \left| \frac{Ans(\tau) - Ans^*}{Ans^*} \right| \leq \epsilon, \\ 0 & \text{otherwise.} \end{cases} \tag{3}$$

where $Ans(\tau)$ is the final answer extracted from trajectory $\tau$, $Ans^*$ is the ground-truth solution, and $\epsilon = 10^{-3}$ in our experiments. We adopt relative error to ensure robustness across problems with different answer scales. We also apply execution-output masking during gradient computation to improve training stability. The final model is referred to as **STORM**.

## 4 Experiments

Our experimental evaluation provides a comprehensive validation of our framework. We first benchmark **STORM** against leading models to establish its state-of-the-art performance. We then conduct extensive ablation and behavioral analyses to dissect the sources of its effectiveness and reveal the mechanisms through which **CALM** reshapes the model's reasoning.

### 4.1 Experimental Setup

**Benchmarks and Datasets.** Our evaluation is conducted on a diverse suite of five benchmarks: NL4Opt (Ramamonjison et al., 2023), MAMO-Easy, MAMO-Complex (Huang et al., 2024), IndustryOR (Huang et al., 2025), and OptMath (Lu et al., 2025). This selection, consistent with prior state-of-the-art studies (Chen et al., 2025), allows us to rigorously test LRM capabilities across a spectrum of difficulty. All training and test data originate from a larger collection of public datasets (Jiang et al., 2024), which we have rigorously partitioned into non-overlapping training and test sets. A comprehensive breakdown of all data sources and our splitting strategy is provided in Appendix E.1.

**Baselines.** We benchmark STORM against a comprehensive set of baselines for a holistic performance evaluation. The comparison includes: (1) **Foundation Models**: GPT-3.5-Turbo, GPT-4 (Achiam et al., 2023) and DeepSeek-V3; (2) **Large Reasoning Models**: DeepSeek-R1-0528 (DeepSeek-AI, 2025) and Qwen3-235B-A22B-Thinking-2507 (Qwen Team, 2025); (3) **Agent-Based Methods**: Chain-of-Experts (Xiao et al., 2023) and OptiMUS (AhmadiTeshnizi et al., 2024); (4) **Learning-Based Methods**: ORLM (Huang et al., 2025), LLMOPT (Jiang et al., 2024), OptMath (Lu et al., 2025) and SIRL (Chen et al., 2025); and (5) crucially, our **Base LRM**, Qwen3-4B-Thinking-2507, which serves as the starting point to directly measure our framework's impact.

**Evaluation Protocol.** We report **pass@1** accuracy as the primary evaluation metric. To address the high variance of greedy decoding in LRMs, as noted in DeepSeek-R1 (DeepSeek-AI, 2025), we follow their recommended evaluation protocol. Specifically, for each problem, we generate 8 independent samples using their specified configuration (temperature=0.6, top-p=0.95). The final **pass@1** score is then reported as the average success rate across these 8 samples. This established method ensures a more robust and reproducible measure of a model's performance. For a fair com-

parison, all LRM-based models are evaluated under this protocol, allowing a maximum of 4 code executions per reasoning trajectory.

**Training Procedure.** For CALM data synthesis, we use Qwen3-4B-Thinking-2507 (Qwen Team, 2025) as the Reasoner and Gemini-2.5-Pro (Comanici et al., 2025) as the Intervener. The curated trajectories are then used in a two-stage training pipeline described in Section 3.3. Our final model, **STORM**, is obtained through this pipeline. Detailed implementations are provided in Appendix E.3.

## 4.2 MAIN RESULTS

**Table 2:** Main results on optimization modeling benchmarks. **Bold** indicates the best performance in each column. Results marked with * are cited from their original papers; all other results are from our own evaluation under a unified protocol. The colored value next to our model's scores indicates the absolute performance gain over its base model.

| Models | Model Size | NL4OPT | MAMO Easy | MAMO Complex | IndustryOR | OptMath | Macro AVG |
|---|---|---|---|---|---|---|---|
| *Baseline Models* | | | | | | | |
| GPT-3.5-Turbo | NA | 78.0* | 79.3* | 33.2* | 21.0* | 15.0* | 45.3* |
| GPT-4 | NA | 89.0* | 87.3* | 49.3* | 33.0* | 16.6* | 55.0* |
| DeepSeek-V3 | 671B | 95.9* | 88.3* | 51.1* | 37.0* | 32.6* | 61.0* |
| DeepSeek-R1-0528 | 671B | 86.6 | 78.8 | 69.1 | 52.5 | **50.6** | 67.5 |
| Qwen3-235B-A22B-Thinking-2507 | 235B | 75.8 | 77.2 | 63.6 | **53.2** | 49.6 | 63.9 |
| *Agent-Based Methods* | | | | | | | |
| Chain-of-Experts | NA | 64.2* | - | - | - | - | - |
| OptiMUS | NA | 78.8* | 77.2* | 43.6* | 31.0* | 20.2* | 49.4* |
| *Learning-Based Methods* | | | | | | | |
| LLMOPT-Qwen2.5-14B | 14B | 80.3* | 89.5* | 44.1* | 29.0* | 12.5* | 51.1* |
| ORLM-LLaMA-3-8B | 8B | 85.7* | 82.3* | 37.4* | 38.0* | 2.6* | 49.2* |
| OptMATH-Qwen2.5-7B | 7B | 94.7* | 86.5* | 51.2* | 20.0* | 24.4* | 55.4* |
| SIRL-Qwen2.5-7B | 7B | **96.3*** | **90.0*** | 62.1* | 33.0* | 29.0* | 62.1* |
| *Our Framework: Transforming a 4B LRM* | | | | | | | |
| Qwen3-4B-Thinking-2507 (Base) | 4B | 85.8 | 73.8 | 46.5 | 46.2 | 33.1 | 57.1 |
| STORM-Qwen3-4B (Ours) | 4B | 93.3 +7.5 | 86.3 +12.5 | **70.3** +23.8 | 50.0 +3.8 | 44.5 +11.4 | **68.9** +11.8 |

We present the main results in Table 2, which demonstrate how our framework transforms a capable LRM into a state-of-the-art optimization modeling expert. We highlight three key findings from our analysis.

First, our method **unlocks a significant leap in performance over the base model**. The initial "calm" adaptation through CALM lays the foundation for STORM to achieve a remarkable gain of **+11.8** absolute points in macro-average accuracy (57.1% to 68.9%), with particularly strong improvements on challenging benchmarks like `MAMO-Complex` (+23.8 points). Second, this enhancement allows our compact 4B model to exhibit **strong parameter efficiency**, achieving performance comparable to the 671B DeepSeek-R1-0528 (68.9% vs. 67.5%) and setting a new state-of-the-art on `MAMO-Complex` (70.3%). Finally, this result **advances the frontier for learning-based methods**, moving beyond the performance benchmarks set by prior works, including the previous SOTA, SIRL (68.9% vs. 62.1%).

These results underscore our central finding: preserving and refining a model's native reasoning patterns can achieve expert-level performance with high parameter efficiency.

## 4.3 ANALYSIS AND ABLATION STUDIES

### 4.3.1 ABLATION STUDY: THE TWO-STAGE LEAP TO SOTA

We analyze the distinct contributions of our two training stages by tracking the performance evolution from the base LRM through SFT and RL, as detailed in Figure 4.

**SFT as a Calibrator.** SFT with CALM-curated data acts as a behavioral calibrator. Unlike direct SFT (Table 1), our soft adaptation avoids performance degradation on complex tasks, yielding a modest gain in macro-average accuracy (57.1% to 58.7%). This stage gently corrects reasoning flaws without overwriting native patterns, laying a stable foundation for subsequent mastery.

**RL as the Accelerator.** Building on this calibrated foundation, the RL stage acts as an accelerator, driving a decisive performance leap. The macro-average accuracy rises sharply from 58.7% to

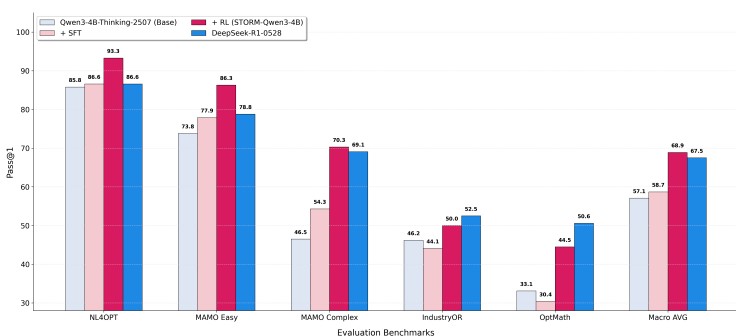

**Figure 4:** Ablation study of our two-stage framework.

68.9%, with the most significant gains on complex reasoning benchmarks. As shown in Figure 4, this "storm" stage propels our 4B model to a level comparable with a 671B LRM, demonstrating a highly parameter-efficient path to expert performance.

### 4.3.2 DECONSTRUCTING CALM: AN INSIDE LOOK AT THE CURATION PROCESS

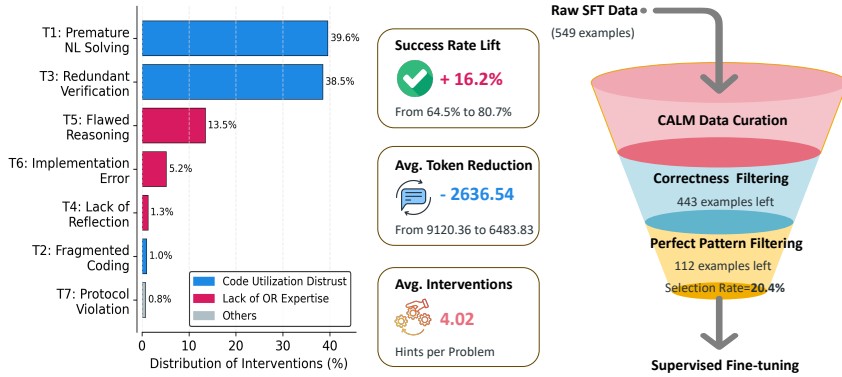

**Figure 5:** The CALM data curation engine.

To understand its mechanics, we decompose the CALM data curation process into three phases as summarized in Figure 5: diagnosing native flaws, refining trajectories via hinting, and filtering the results into a high-quality SFT dataset.

**Diagnosis of Native Flaws.** The diagnosis phase identifies failure modes in the base LRM's initial trajectories. The distribution of interventions (Figure 5, left) reveals two dominant flaw categories: *Code Utilization Distrust* and *Lack of OR Expertise*. Consistent with our analysis in Section 2.3, the former is more prevalent on the low-to-medium difficulty problems common in our SFT set.

**Refinement via Lightweight Hinting.** The refinement phase uses an iterative hinting loop to correct flawed trajectories. As shown in Figure 5 (middle), this lightweight process, with minimal interventions per problem, significantly boosts the success rate while simultaneously reducing response length. This demonstrates that targeted guidance can enhance both correctness and conciseness.

**Filtering of "Golden" Trajectories.** Finally, the filtering phase ensures only the highest-quality expert demonstrations are used for training. Our rigorous filtering funnel (Figure 5, right) is highly selective, retaining only trajectories that are both correct and deemed flawless by the Intervener, which guarantees the purity of the supervision signal for the SFT stage.

### 4.3.3 BEHAVIORAL EVOLUTION: HOW CALM SHAPES REASONING

To understand *why* CALM-SFT makes reinforcement learning more efficient, we conduct a controlled experiment. We compare two models starting from the same base LRM: **RL with CALM**, fine-tuned on our curated "golden" trajectories, and a control model, **RL without CALM**, fine-tuned on the original unguided reasoning flows. This design isolates the effect of the initial SFT data quality on the RL process. The detailed setup is provided in Appendix E.4.

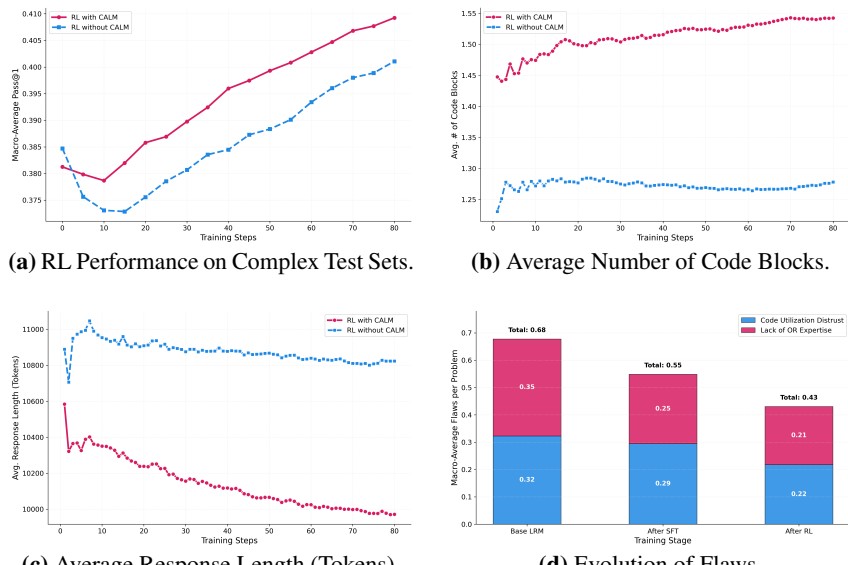

**(a)** RL Performance on Complex Test Sets.

**(b)** Average Number of Code Blocks.

**(c)** Average Response Length (Tokens).

**(d)** Evolution of Flaws.

**Figure 6:** Behavioral evolution analysis.

**CALM as a Catalyst for Sample-Efficient RL.** Figure 6a shows that starting RL from high-quality reasoning patterns has a profound impact. The RL with CALM model exhibits a steeper and more stable learning curve, achieving a noticeably higher performance ceiling within the same computational budget. In contrast, the control model learns far more slowly and shows no indication of closing the performance gap. This confirms that SFT on CALM trajectories provides a strong inductive bias, acting as a catalyst that makes subsequent RL far more sample-efficient.

**A Shift Toward Computation-Driven Reasoning.** This efficiency is explained by consistent behavioral changes, as shown in Figures 6b and 6c. The RL with CALM model progressively increases its use of code blocks while reducing average response length. This reflects a shift toward expert-like behavior: replacing verbose natural language calculations with concise and reliable code execution. The control model, lacking this guidance, remains verbose and less computation-driven.

**The Two-Stage Healing Process.** Finally, Figure 6d reveals a complementary "healing process." The SFT stage shows a larger impact on reducing *Lack of OR Expertise*, while the subsequent RL stage is more effective at reducing *Code Utilization Distrust*. Together, these stages synergistically transform the LRM into a specialized optimization modeler. A per-benchmark breakdown of this evolution is provided in Appendix F.

## 5  CONCLUSION

This work introduces **CALM**, a lightweight framework for adapting Large Reasoning Models (LRMs) to optimization modeling. By aligning targeted interventions with specific reasoning flaws, CALM preserves native reasoning capabilities while improving optimization modeling accuracy. Our two-stage training pipeline — combining hint-guided supervised fine-tuning with reinforcement learning — transforms a compact LRM into **STORM**, which achieves state-of-the-art performance across diverse benchmarks. These results demonstrate the effectiveness of minimally invasive, reasoning-aligned adaptation for domain specialization. A promising direction for future work is to extend **STORM** to broader optimization modeling agent frameworks, such as OptiMUS (AhmadiTeshnizi et al., 2024).

## REPRODUCIBILITY STATEMENT

To ensure the reproducibility of our findings, we provide comprehensive details of our experimental setup and plan to release our code and models.

- **Code and Models:** We plan to release our model and code.
- **Data Details:** Appendix E.1 provides a detailed breakdown of all public benchmarks used, including their sources, descriptions, and our specific data splitting strategy.
- **Training Details:** Comprehensive hyperparameters and implementation details for all training stages are provided in Appendix E.3.
- **Computing Infrastructure:** The hardware used for all experiments is also described in Appendix E.3.

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

## A    RELATED WORK

**From Non-Reflective to Reflective OR Modeling.** The application of LLMs to OR is undergoing a fundamental paradigm shift. Early learning-based methods, including ORLM (Huang et al., 2025), LLMOPT (Jiang et al., 2024), and SIRL (Chen et al., 2025), treated modeling as a non-reflective generation task, training models to produce a complete solution in a single, static pass. This approach, however, is misaligned with modern LRMs (Qwen Team, 2025; DeepSeek-AI, 2025), which possess powerful native reasoning patterns for iterative and adaptive problem-solving. Our work embraces this shift, aiming to preserve and guide the LRM's inherent capacity for reflective generation.

**Hint-based Reasoning Adaptation.** Injecting guidance into a model's reasoning is a promising adaptation technique. Concurrent works like START (Li et al., 2025b) use pre-defined, static hint libraries to encourage code use, while CoRT (Li et al., 2025a) relies on manual, human-in-the-loop annotation for its Hint Engineering, limiting scalability. In contrast, our CALM framework introduces a fully automated and dynamic "Reasoner-Intervener" pattern. It moves beyond static libraries and manual oversight by enabling an expert model to detect flaws and inject tailored hints, providing scalable, process-level correction that respects the LRM's native reasoning.

## B    ILLUSTRATION: INPUT–OUTPUT STRUCTURE OF TRADITIONAL LLMS
## FOR OPTIMIZATION PROBLEMS

---

**Example: Optimization Problem in Natural Language and Its Formalization**

**Input (Natural-Language Problem).** A company must transport 25 tons of cargo using trucks, airplanes, or ships. Per-ton costs are \$100, \$120, and \$130, with capacities 10, 20, and 30 tons. Trucks and ships cannot be chosen together. The goal is to minimize the total cost while satisfying the demand.

- - - - - - - - - - - - - - - - - - - - - - - - - - - - - - - - - - - - - - - - -

**Output (Mathematical Model and Solver Code).**
*Variables.*

- $x_1, x_2, x_3 \in \{0, 1\}$: binary variables indicating whether trucks, airplanes, and ships are selected.

- $y_1, y_2, y_3 \geq 0$: transported volumes (tons) by each mode.

*Objective and Constraints.*

$$\min 100y_1 + 120y_2 + 130y_3$$
$$\text{s.t.} \quad x_1 + x_2 + x_3 \geq 1 \tag{4}$$
$$y_1 \leq 10x_1, \quad y_2 \leq 20x_2, \quad y_3 \leq 30x_3 \tag{5}$$
$$x_1 + x_3 \leq 1 \tag{6}$$
$$y_1 + y_2 + y_3 \geq 25 \tag{7}$$

*Program (PuLP).*

```python
from pulp import LpProblem, LpMinimize, LpVariable, LpBinary, lpSum,
    PULP_CBC_CMD, value

# Data
costs = {"trucks":100, "airplanes":120, "ships":130}
caps  = {"trucks":10,  "airplanes":20,  "ships":30}
demand = 25

# Model
m = LpProblem("Transportation", LpMinimize)
x = {k: LpVariable(f"x_{k}", 0, 1, cat=LpBinary) for k in costs}
y = {k: LpVariable(f"y_{k}", 0) for k in costs}

# Objective
m += lpSum(costs[k]*y[k] for k in costs)
```

```
15
16  # Constraints
17  m += lpSum(x[k] for k in costs) >= 1
18  for k in costs:
19      m += y[k] <= caps[k]*x[k]
20  m += x["trucks"] + x["ships"] <= 1
21  m += lpSum(y[k] for k in costs) >= demand
22
23  # Solve
24  m.solve(PULP_CBC_CMD(msg=False))
25  print("Objective:", value(m.objective))
26  for k in costs:
27      print(f"{k}: x={value(x[k])}, y={value(y[k])}")
```

## C  PROTOCOL FOR HUMAN-IN-THE-LOOP FLAW TAXONOMY CREATION

This section details the rigorous, multi-stage protocol our team of four human experts (graduate students with OR and STEM backgrounds) followed to establish the seven-flaw taxonomy presented in Section 2.3. The goal was to move from unstructured observations to a systematic and reproducible classification of errors.

**Stage 1: Initial Data Generation and Independent Annotation.** A base LRM (Qwen3-4B-Thinking-2507) was used to generate solutions for a diverse set of 50 problems selected to cover a range of difficulties and types from our benchmark suite. Each of the four annotators independently reviewed these same 50 responses. For each response, they performed an open-ended analysis, identifying and documenting any perceived reasoning errors. Annotators were instructed to assign a descriptive tag (e.g., "manual-calculation-error," "missed-integer-var") and provide a brief textual justification for each identified flaw. This initial stage resulted in four independent sets of annotations, containing a rich but unstructured collection of observed errors.

**Stage 2: Collaborative Clustering and Taxonomy Refinement.** The team then engaged in a collaborative session to synthesize the independent findings. The process was as follows:

1. **Merging**: All unique error tags and justifications from the four annotators were collected into a single master list.

2. **Affinity Clustering**: The team collectively grouped semantically similar tags into higher-level clusters. For example, tags like "manual-calculation-error," "avoids-solver," and "solves-by-hand" were grouped into a cluster that would later become "Premature NL Solving."

3. **Definition and Refinement**: For each cluster, the team collaboratively wrote a precise, operational definition for the flaw type it represented. This process involved several rounds of discussion to ensure the definitions were mutually exclusive and collectively exhaustive for the observed phenomena. Any ambiguous or overlapping clusters were either merged or further refined.

This iterative process led to the convergence on the seven distinct and recurring flaw types detailed in Appendix D. This human-in-the-loop methodology ensures that our taxonomy is grounded in empirical observation and expert consensus.

## D  TRIGGERS TYPE

As detailed in our protocol (Appendix C), our analysis identified seven recurring flaw types. Six of these are classified as *substantive reasoning flaws* as they represent fundamental errors in the problem-solving process. The seventh, *Protocol Violation*, is classified as a *procedural error* as it relates only to output formatting. Our main analysis in the paper focuses on the six substantive flaws. The definitions for all seven triggers are as follows:

- **[Trigger 1] Premature NL Solving**: After formulating the mathematical model, the LRM starts solving it manually with natural language instead of immediately writing solver code.

- **[Trigger 2] Fragmented Coding**: The LRM writes small, non-executable, or multiple solver-running code blocks instead of a single, comprehensive one.

- **[Trigger 3] Redundant Manual Verification**: After a code output, the LRM manually re-calculates the exact numerical results that were already provided by the solver.

- **[Trigger 4] Lack of Sanity Check/Reflection**: The LRM gets a correct code output but proceeds directly to the final answer without any high-level reflection on the result's plausibility.

- **[Trigger 5] Flawed Reasoning or Modeling**: The LRM's logic is flawed, leading to an incorrect answer. This includes semantic misunderstanding, a wrong mathematical model, or missing constraints (e.g., integers).

- **[Trigger 6] Implementation Error**: The mathematical model is correct, but the code is buggy or does not faithfully represent the model, leading to an incorrect answer.

- **[Trigger 7] Protocol Violation**: The LRM violates a clear instruction, especially regarding the final boxing requirement.

Here, triggers 1-3 exemplify **Code Utilization Distrust**, pinpointing behaviors such as solving problems with natural language instead of code or engaging in inefficient coding practices. Triggers 4-6 are indicators of a **Lack of OR Expertise**, covering fundamental errors in modeling, logical reasoning, and code implementation. A final trigger, Protocol Violation (Trigger 7), serves as a procedural check to ensure the model adheres to specific output formatting instructions. See Table 3 for examples.

**Table 3:** This table illustrates seven common LRM error patterns ('triggers'), showing the original error (red) and analysis of the errors. These triggers include: **(1)Premature NL Solving**, an attempt at manual calculation instead of coding; **(2)Fragmented Coding**, writing separate small code blocks; **(3) Redundant Manual Verification**, unnecessarily re-calculating a solver's result; **(4) Lack of Sanity Check**, failing to reflect on a solution's plausibility; **(5) Flawed Reasoning or Modeling**, formulating an incorrect mathematical model; **(6) Implementation Error**, correctly modeling a question but incorrectly coding a correct model; and **(7)Protocol Violation**, ignoring explicit instructions and embedding the boxed number within a sentence.

---

**Error Type [Trigger 1] Premature NL Solving**: ... I notice that there is only 5 combinations and this is an easy task, so I can just try them all first without writing python code. ... Alternatively, Worker III → B (4), Worker IV → D (3), Worker I → A (9), Worker V → C (7). Total: 4 + 3 + 9 +7 = 23. That's worse. ...
**Analysis**: It incorrectly assumed the problem was a simple task involving only five combinations, thus obviating the need for Python code. The appropriate methodology would be to first construct a mathematical model and then employ a solver, accessed via code, to obtain the solution.

---

**Error Type [Trigger 2] Fragmented Coding**: ... Re-calculating the derivative and solving by hand is unnecessary ... I'll calculate the ratio 'h/r' directly from the output. I can even use another small code block to be precise.

```python
# Calculate the ratio h/r to determine the shape
ratio = h / optimal_r
print(f"Height to radius ratio: {ratio.evalf()}")
```

```output
...
NameError: name 'h' is not defined
```

**Analysis**: This is fragmented coding and will fail because the variables from the first block are not available in the second. The ideal workflow is to perform all related calculations in a single, self-contained solver script.

---

**Error Type [Trigger 3] Redundant Manual Verification**: ... So the code is correct. (But wait, another way: Let's see if there's a possible solution with lower total flavouring. For example, $M = 1, O = 2$ gives 10.) ...

**Analysis**:After correctly executing the solver and getting the optimal result, the LRM shows a lack of trust in the tool. It proceeds to manually verify the constraints (which is acceptable) but then attempts to re-solve the problem by hand.

---

**Error Type [Trigger 4] Lack of Sanity Check/Reflection**: ...

```output
Total cost: 24.00
Food 1: 0.0
Food 2: 8.0
...
Food 9: 4.0
```

The output shows that the **optimal solution** uses:

- **8 units of Tofu**

- **4 units of Quinoa**

This combination meets all the nutritional requirements at a **minimum total cost of 24.00**.

**Analysis**: The LRM correctly executes the solver and gets the optimal solution. However, it moves directly from stating the decision variables (8 Tofu, 4 Quinoa) to declaring the final answer without performing a brief sanity check. An expert would quickly verify that this proposed solution indeed satisfies the nutritional constraints and matches the calculated cost.

---

**Error Type [Trigger 5] Flawed Reasoning or Modeling**:: ... But maybe in this problem, we can treat them as continuous variables (since the problem might be designed for linear programming without integer constraints) ...

**Analysis**: In the context of this problem, the units of the products must be integers and cannot be continuous. The LRM incorrectly treated continuous variables as integer variables during the modeling process, which led to a flawed model formulation.

---

**Error Type [Trigger 6] Implementation Error**:: ...

**Constraints**:
1. **Demand Constraint**: For each store $j$, $\sum_{i=1}^{6} y_{ij} \geq \text{demand}_j$
2. **Capacity Constraint**: For each DC $i$, $\sum_{j=1}^{9} y_{ij} \leq \text{capacity}_i$
3. **Non-negativity**: $y_{ij} \geq 0$ ...

```python
...
# Create flow variables y[i][j] for each DC i and store j
 y = [[pulp.LpVariable(f'y_{i}_{j}', cat='Continuous') for j in range(9)] for i in range(6)]
...
```

```output
 Status: Unbounded
 Total cost: 16417.0
```

...

**Analysis**: Although the LRM was correctly established in the modeling phase with constraints ensuring the non-negativity of all variables $y_{ij}$, this requirement was overlooked during implementation, where the code failed to set a lower bound of zero for $y_{ij}$.

---

**Error Type [Trigger 7] Protocol Violation**: ...

**Final Answer**

...I'll now summarize the findings and box the final answer. The optimal solution uses 97.01 square feet of sunflowers and 0 square feet of roses, yielding a maximum profit of $\boxed{43656.72}$ .

> **Analysis**: The LRM's final answer formulation violates the instructions. It embeds the boxed number within a sentence, whereas the protocol requires the box to contain only the final numerical answer and be separate from the summary text.

# E  EXPERIMENTAL DETAILS APPENDIX

## E.1  BENCHMARK DATASETS AND SPLITTING STRATEGY

Our study utilizes a broad range of public benchmarks Jiang et al. (2024) for training and evaluation. To ensure a rigorous and unbiased experimental design, we randomly partitioned all available data from eight sources into non-overlapping training (SFT and RL) and test sets. Table 4 provides a comprehensive overview of these sources, their original sizes, and our final partitioning.

While our main evaluation in the paper focuses on five key benchmarks to ensure direct comparability with prior state-of-the-art work (Chen et al., 2025), we provide test splits for all datasets to facilitate future research.

**Table 4:** Comprehensive overview of benchmark datasets and our rigorous splitting into non-overlapping SFT, RL, and Test sets.

| Data Source | | | Data Partitioning | | |
|---|---|---|---|---|---|
| **Benchmark** | **Description** | **Original Size** | **SFT Set** | **RL Set** | **Test Set** |
| NL4Opt | NeurIPS 2022 competition data, focusing on LP formulation. | 46 | 8 | 8 | 30 |
| MAMO-Easy | High-school level MILP problems for fundamental modeling. | 650 | 200 | 350 | 100 |
| MAMO-Complex | Undergraduate-level MILP/LP problems with intricate structures. | 211 | 55 | 56 | 100 |
| IndustryOR | Real-world industrial problems across diverse sectors and types. | 100 | 6 | 12 | 80 |
| OptMath | Challenging mathematical optimization problems for advanced reasoning. | 166 | 30 | 36 | 100 |
| OptiBench | A collection of various optimization problems. | 607 | 250 | 257 | 100 |
| ComplexOR | Complex OR problems from academic and industrial scenarios. | 18 | 0 | 0 | 18 |
| NLP4LP | LP problems sourced from optimization textbooks and lecture notes. | 12 | 0 | 0 | 12 |

## E.2  IMPLEMENTATION DETAILS FOR THE PILOT STUDY

This section provides the specific implementation details for the pilot study discussed in Section 2.2.

- **Base Large Reasoning Model (LRM):** The LRM used in this study was **Qwen3-4B-Thinking-2507**, a powerful open-source model known for its strong multi-step reasoning capabilities.
- **Non-reflective Dataset:** We used **OR-Instruct-3K** (Huang et al., 2025), a widely-recognized dataset in the field. It consists of 3,000 problem-solution pairs and is representative of the non-reflective data generation paradigm.
- **Training Procedure:** The base LRM was fine-tuned using a standard supervised fine-tuning (SFT) objective. The training utilized the same set of hyperparameters as our main SFT stage, which are detailed in Table 5.

## E.3  IMPLEMENTATION DETAILS FOR THE CALM & STORM FRAMEWORK

This section provides a comprehensive overview of the implementation details for our entire framework, including the computing infrastructure, the CALM data curation process, and the two-stage training pipeline.

**Computing Infrastructure.**  All experiments were conducted on a cluster of four nodes, each equipped with 8x NVIDIA H800 (80GB) GPUs.

**CALM Data Curation.** The expert-aligned trajectories for SFT were generated using our `CALM` framework with the following configuration:

- **Reasoner Model:** `Qwen3-4B-Thinking-2507`.
- **Intervener Model:** `Gemini-2.5-Pro`.
- **Process Control:** The iterative hinting loop was run for a maximum of $N = 5$ interventions per problem. An "intervention" consists of the Intervener identifying a flaw, injecting a hint, and the Reasoner regenerating the trajectory from that point. This limit serves as a practical safeguard to prevent excessively long or unproductive correction cycles. If a trajectory remains flawed after 5 interventions, it is discarded and not considered for the final SFT dataset.
- **Reasoner Generation Parameters:** Temperature set to 0.6, top-p to 0.95. Max response length was 16384 tokens with a maximum of 4 code executions per turn.
- **Intervener Generation Parameters:** Temperature set to 1.0 and top-p to 0.95 to encourage diverse analytical feedback.

**Stage 1: Supervised Fine-Tuning (SFT).** The SFT stage used the 112 "golden" trajectories curated by the `CALM` process.

- **Base Model:** `Qwen3-4B-Thinking-2507`.
- **Optimizer:** AdamW.
- **Key Hyperparameters:** Summarized in Table 5.
- **Framework:** DeepSpeed Stage 3 with bf16 precision.

Table 5: Key hyperparameters for the supervised fine-tuning (SFT) stage.

| Hyperparameter | Value |
|---|---|
| Learning Rate | 1e-5 |
| LR Scheduler | Cosine |
| Warmup Ratio | 0.1 |
| Total Batch Size | 8 |
| Number of Epochs | 3 |
| Max Sequence Length | 22000 |

**Stage 2: Reinforcement Learning (RL).** The RL stage commenced from the final checkpoint of the SFT model, using the following setup:

- **Algorithm:** Group Relative Policy Optimization (GRPO) via the Verl framework (Sheng et al., 2024).
- **Key Hyperparameters:** Detailed in Table 6.

### E.4 IMPLEMENTATION DETAILS FOR THE CONTROLLED EXPERIMENT

This section details the setup for the controlled experiment presented in Section 4.3.3, which was designed to isolate the impact of the initial SFT data quality on RL dynamics. The experiment involved a direct comparison between our main model, RL with CALM, and a control model, RL without CALM. To ensure a rigorous comparison, the control model's setup was designed to mirror the main model's in every aspect except for the SFT data.

**SFT Data.** The control model was fine-tuned on the 112 *original, unguided* reasoning trajectories corresponding to the same problems used for the main model's SFT stage.

**Hyperparameters.** To maintain a controlled environment, the hyperparameters for the control model's SFT and RL stages were kept identical to those of our main model. Due to computational resource constraints, the RL training for this specific comparative analysis was conducted for 30 epochs. The complete list of hyperparameters for the control model is provided in Appendix E.3 for full transparency.

Table 6: Key hyperparameters for the Reinforcement Learning stage.

| Hyperparameter | Value |
|---|---|
| **General** | |
| Start Model Checkpoint | Final from supervised fine-tuning |
| Learning Rate | 1e-6 |
| Total Epochs | 100 |
| Train Batch Size | 64 |
| PPO Mini-batch Size | 64 |
| KL Loss | Disabled |
| **Rollout Configuration** | |
| Samples per Prompt (N) | 8 |
| Temperature | 0.6 |
| Max Prompt Length | 3000 |
| Max Response Length | 16384 |
| Max Code Execution per Rollout | 4 |

## F    DETAILED BREAKDOWN OF FLAW FREQUENCY EVOLUTION

In Section 4.3.3 of the main text, we presented the macro-average trend of flaw frequency reduction. To provide a more granular view, Figure 7 presents a detailed, per-benchmark breakdown of this evolution.

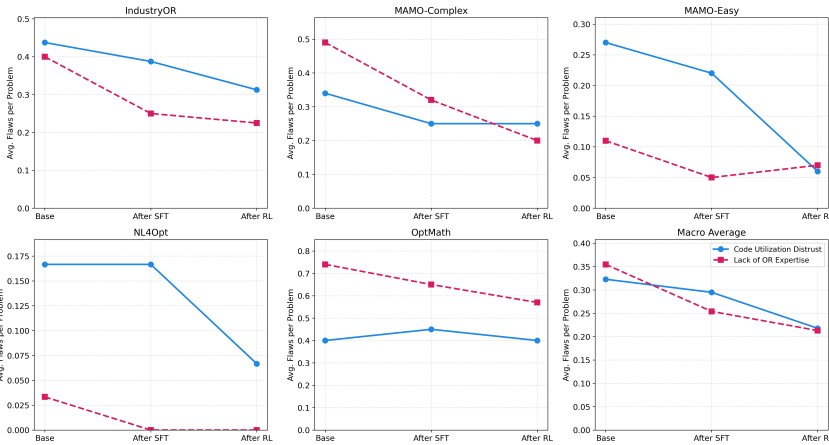

**Figure 7:** A per-benchmark breakdown of the evolution of flaw frequencies. Each subplot shows the average number of flaws per problem for the two main categories across the three training stages: Base LRM, After SFT, and After RL. The 'Macro Average' plot (bottom right) summarizes the general trend.

The six-panel figure illustrates the change in frequency for the two primary flaw categories—*Code Utilization Distrust* (blue solid line) and *Lack of OR Expertise* (red dashed line)—at each training stage.

A detailed analysis of the trends reveals the complementary roles of our two-stage approach:

- **Stage 1 (SFT): Broad-Spectrum Correction.** The supervised fine-tuning stage initiates a significant reduction in both types of flaws across almost all benchmarks. Notably, we observe a substantial drop in the red line (*Lack of OR Expertise*) during this phase (e.g., in `IndustryOR` and `MAMO-Complex`). This suggests that exposing the LRM to high-quality, expert-aligned reasoning trajectories in the CALM dataset provides strong initial guidance, helping it to correct fundamental modeling errors and adopt more expert-like problem formulations. The blue line (*Code Utilization Distrust*) also shows a general downward trend, indicating that the model begins to learn more efficient code-use habits.

- **Stage 2 (RL): Targeted Refinement and Mastery.** Building upon the foundation laid by SFT, the reinforcement learning stage continues to refine the model's skills. The RL phase consistently drives down the remaining flaws of both types, pushing the error rates to their lowest levels. This stage allows the model to move beyond simple imitation and achieve a deeper, more robust mastery of both domain knowledge and code use through trial-and-error exploration.

This per-benchmark analysis reinforces our central claim: the two-stage pipeline works synergistically. SFT provides a strong initial correction across the board, and RL builds upon this to achieve a state of expert-level proficiency.

# G COMPARISON OF REASONING PARADIGM

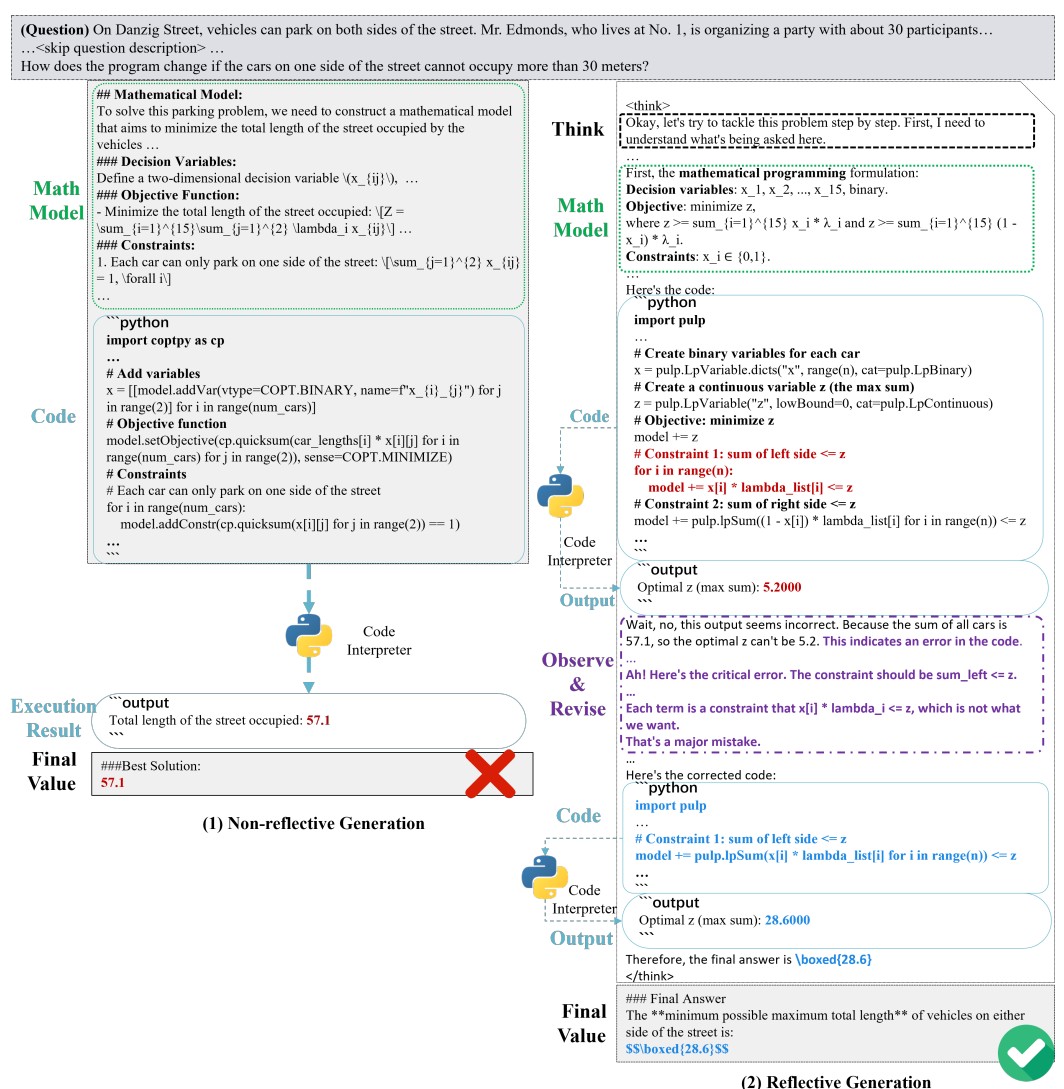

**Figure 8:** An illustrative example comparing the Non-reflective Generation (left) and Reflective Generation (right) paradigms on a vehicle parking optimization problem

Figure 8 demonstrates the practical differences between the two reasoning paradigms using a parking optimization task. The Non-reflective Generation approach (left) formulates a mathematical model and writes the complete code in a single step. However, a subtle error in one of the constraints leads to a logically incorrect final answer. Due to its non-reflective pattern, the model is unable to detect or correct this error.

In contrast, the reflective generation approach (right) showcases an iterative refinement process. The model initially generates code that also contains an error, leading to an implausible output. By observing this solver output, the model identifies the flaw in its reasoning. It then autonomously corrects the constraint in the code and re-executes it, successfully arriving at the correct optimal value. This case clearly highlights the advantage of the reflective paradigm: its ability to leverage execution feedback for self-correction and robust problem-solving.

# H INTERVENTION PROCESS FOR SPECIFIC FLAWS

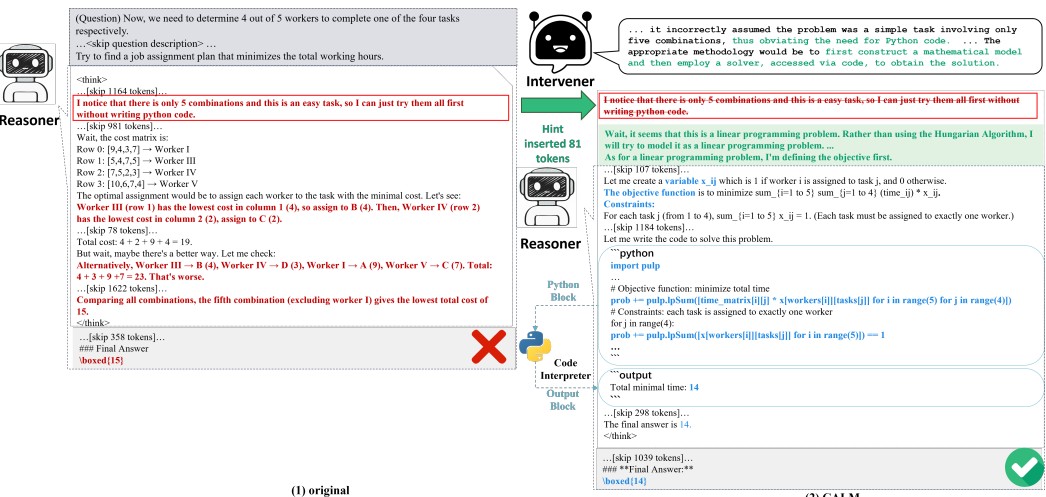

**Figure 9:** A representative example of the *Code Utilization Distrust* flaw. (1) The model exclusively relies on verbal reasoning and fails to utilize the solver, leading to an incorrect answer. (2) In contrast, the reasoning process guided by **CLAM** successfully employs the solver to reach the correct solution.

# I ILLUSTRATIVE CASE STUDY OF THE CALM FRAMEWORK

To complement our quantitative findings, we provide a case study illustrating how CALM works in practice. Figure 10 traces a multi-turn "Reasoner–Intervener" interaction. The initial trajectory begins with a *Lack of OR Expertise* error, mistaking an Integer Linear Program (ILP) for a standard LP. Even after correction, a secondary issue of *Code Utilization Distrust* emerges. This case demonstrates how a sequence of lightweight, targeted hints can progressively guide the Reasoner through distinct failure modes, ultimately yielding a correct, expert-aligned solution without further intervention.

---

**Reasoning** ($\tau^{(0)}$): ... But maybe in this problem, we can treat them as continuous variables (since the problem might be designed for linear programming without integer constraints) ... *Answer:* 798.04 (Wrong!)

---

**Intervention** ($i = 0$, *Trigger 5 – "Flawed Reasoning or Modeling" detected* ): I've correctly identified the ambiguity around continuous vs. integer variables ... This changes the problem from a Linear Program (LP) to an Integer Linear Program (ILP). ...

---

**Reasoning** ($\tau^{(1)}$): ... So the code is correct. (But wait, another way: Let's see if there's a possible solution with lower total flavouring. For example, $M = 1$, $O = 2$ gives 10.) ... 10 (Correct!)

---

**Intervention** ($i = 1$, *Trigger 3 – "Redundant Manual Verification" detected* ): Okay, the solver returned M=1, O=2, with a total flavouring of 10. I should trust the solver's optimality result. ... I'll proceed to structure the final answer.

---

**Reasoning** ($\tau^{(2)}$): The solver confirms that the **optimal solution** is: 1 order of **matcha ice cream**; 2 orders of **orange sorbet**... 10 (Correct!)

---

**Intervention** ($i = 2$, *Trigger 3 – "Redundant Manual Verification" detected* ): ... This result is logical, and I can now confidently format the final answer.

---

**Reasoning** ($\tau^{(3)}$): (correct reasoning without any trigger detected)... 10 (Correct!)

---

**Intervention** ($i = 3$, *"NO INTERVENTION"* )

---

**Figure 10:** An illustrative case study of the iterative "Reasoner–Intervener" collaboration pattern, where targeted hints progressively correct a flawed reasoning trajectory. Here, red represents the error and blue represents the correction of the Intervener.

### I.1 PROMPT TEMPLATES

The effectiveness of our framework relies on carefully designed prompts for both the Reasoner's initial task and the Intervener's supervisory role.

**Initial Prompt for the Reasoner.** The Reasoner is initiated with a detailed prompt that outlines the task, reasoning guidelines, tool usage protocols, and the required final answer format. The full template is provided below.

---

**Prompt for Reasoner**

```
Given a mathematical problem, follow the instructions below to
↪  solve it.

\#\#\# Instructions:

When solving mathematical problems, you should leverage both
↪  natural language reasoning and Python code execution. Your
↪  goal is to provide clear, detailed explanations while
↪  utilizing Python to perform complex calculations. Follow
↪  these guidelines to ensure a coherent and effective
↪  response:

1.  **Natural Language Reasoning:**
    -   Provide comprehensive, step-by-step explanations of your
    ↪  thought process.
    -   Formulate your plan BEFORE writing code. Explain what
    ↪  you are about to do and why.

2.  **Code Execution Rules:**
    -   **Purpose:** Each Python code block must be a complete,
    ↪  self-contained script that executes a single, logical
    ↪  step of your plan.
    -   **Output:** The SOLE mechanism for displaying results is
    ↪  the `print()` function. The purpose of a code block is
    ↪  to compute a value or set of values and explicitly
    ↪  `print()` them for the subsequent `output` block.
    -   **Structure:** Each block must contain all necessary
    ↪  imports and setups. The code must be directly
    ↪  executable. Avoid any boilerplate like `if \_\_name\_\_
    ↪  == '\_\_main\_\_':`.

3.  **Recommended Toolkit & Best Practices:**
    -   To ensure reliability and environment compatibility,
    ↪  **you must prioritize using the following libraries**
    ↪  for their respective tasks.
    -   For **symbolic mathematics**: use `sympy`.
    -   For **numerical operations**: use `numpy`.
    -   For **scientific computing**: use `scipy`.
    -   For **optimization problems**: use `pulp`.

4.  **Solution Verification and Final Answer:**
    A. **Code Output for Verification:** To ensure your
    ↪  reasoning is transparent and verifiable, your **final
    ↪  code block** should print all key results needed for the
    ↪  solution. For optimization problems, this typically
    ↪  includes:
        *   The optimal objective function value.
```

```
          *    The values of the main decision variables.

    C. **Final Answer Formulation:**
          *    **Full Solution Description:** Briefly summarize
          ↪    your findings, referencing the key values printed by
          ↪    your code.
          *    **Final Answer Boxing:** The final step is to put
          ↪    the **single numerical answer** to the main question
          ↪    inside `\\boxed{}`.
             - **Content:** The box should contain **only the
             ↪    number**, without any units, currency signs, or
             ↪    explanatory text.
             - **Example (Correct):** `\\boxed{1234}` or
             ↪    `\\boxed{1234.37}`
             - **Example (Incorrect):** `\\boxed{Total cost is
             ↪    \$1234.0}`

\#\#\# Problem:
{problem_text}
```

**Prompt for the Intervener.** The Intervener is guided by a meta-prompt that defines its role, the ideal expert workflow, and the specific 'Deviation Triggers' it should look for. This prompt is crucial for the automated and targeted nature of our hinting process. The full template is provided below.

### Prompt for Intervener

```
\#\#\# CONTEXT AND GOAL
You are an expert Operations Research (OR) engineer and an LLM
 ↪  Reasoning Pattern Analyst. Your mission is to assist in
 ↪  generating high-quality training data for fine-tuning Large
 ↪  Reasoning Models (LRMs).

The ultimate goal is to adapt an LRM's native reasoning pattern
 ↪  (which is heavily reliant on long-form natural language) to
 ↪  better emulate the iterative workflow of a human OR expert.
 ↪  The ideal expert workflow is a cycle of: **1. Understand \&
 ↪  Model -> 2. Code Solver -> 3. Execute \& Observe -> 4.
 ↪  Reflect \& Debug -> (Repeat)**.

Your specific task is to analyze a given LRM response and, if it
 ↪  deviates from this ideal workflow, insert a strategic hint
 ↪  to guide it back on track. This process, called "Auto Hint
 ↪  Engineering," creates a more efficient and robust reasoning
 ↪  trace for training.

\#\#\# INSTRUCTIONS
1. First, carefully review the `TASK_DEFINITION` which contains
 ↪  the original problem and instructions given to the LRM.
2. Next, analyze the provided `LLM_RESPONSE_TO_REFINE`.
3. Identify the **first point** of deviation based on the
 ↪  triggers defined below.
4. If a deviation is found, your output MUST be structured
 ↪  using the custom tags `<action>`, `<trigger_type>`,
 ↪  `<analysis>`, `<target_text>`, and `<hint_to_insert>`. The
 ↪  action should be "REPLACE_AND_CONTINUE".
```

```
5.  The `<target_text>` tag should contain the exact, unique,
↪   and contiguous block of text from the original response that
↪   needs to be replaced.
6.  The `<hint_to_insert>` tag should contain the new hint
↪   you've crafted according to the principles below.
7.  If the response is ideal, your output should simply be
↪   `<action>NO_INTERVENTION</action>`.

### DEVIATION TRIGGERS
*   **Trigger 1: Premature NL Solving:** After formulating the
↪   mathematical model, the LRM starts solving it manually with
↪   natural language instead of immediately writing solver code.
*   **Trigger 2: Fragmented Coding:** The LRM writes small,
↪   non-executable, or multiple solver-running code blocks
↪   instead of a single, comprehensive one.
*   **Trigger 3: Redundant Manual Verification:** After a code
↪   output, the LRM manually re-calculates the exact numerical
↪   results that were already provided by the solver.
*   **Trigger 4: Lack of Sanity Check/Reflection:** The LRM gets
↪   a correct code output but proceeds directly to the final
↪   answer without any high-level reflection on the result's
↪   plausibility.
*   **Trigger 5: Flawed Reasoning or Modeling:** The LRM's logic
↪   is flawed, leading to an incorrect answer. This includes
↪   semantic misunderstanding, a wrong mathematical model, or
↪   missing constraints (e.g., integers).
*   **Trigger 6: Implementation Error:** The mathematical model
↪   is correct, but the code is buggy or does not faithfully
↪   represent the model, leading to an incorrect answer.
*   **Trigger 7: Protocol Violation:** The LRM violates a clear
↪   instruction, especially regarding the final boxing
↪   requirement.

\#\#\# HINT PRINCIPLES (to guide your hint creation)
*   **Be a Guide, Not a Dictator:** Use a first-person,
↪   reflective tone (e.g., "I see, a better way would be...",
↪   "Okay, now I should...").
*   **Encourage Action:** Frame the hint to prompt a specific,
↪   desirable next action.
*   **[FOR TRIGGERS 1 \& 2] Force Code Generation:** End your
↪   hint with `\n\n\`\`\`python` to strongly encourage immediate
↪   and complete code writing.
    *   *Example:* "The model is fully formulated. The best next
    ↪   step is to implement this using `pulp` to get an exact
    ↪   solution.\n\n\`\`\`python"
*   **[FOR TRIGGER 3] Promote Trust in Tools:** Guide the LRM
↪   away from redundant calculation and towards interpretation.
    *   *Example:* "The solver has already provided the optimal
    ↪   values. Re-calculating them manually is unnecessary. I
    ↪   should now focus on interpreting the solution."
*   **[FOR TRIGGER 4] Encourage Sanity Checks:** Gently guide
↪   the LRM to perform a brief, high-level sanity check. The
↪   goal is to cultivate a habit of reflection, not to force a
↪   rigid process.
```

* **Hint for Trigger 4 (Lack of Reflection):** "The solver
  ↪ returned an optimal cost of \$392,760. Before I finalize
  ↪ the answer, it's a good practice to quickly reflect on
  ↪ this. Given the high fixed costs of the distribution
  ↪ centers, this value seems to be in a reasonable range.
  ↪ This gives me confidence in the result. Now, I'll
  ↪ proceed to format the final solution."
* **[Alternate Hint with Code-Assisted Check]:** "The
  ↪ solver returned an optimal cost of \$392,760. That seems
  ↪ plausible. To build more confidence, I could write a
  ↪ quick script to explore a simplified scenario, like
  ↪ checking the cost if I only open the three cheapest
  ↪ centers. This will help verify my
  ↪ understanding.\n\n\`\`\`python"

* **[FOR TRIGGERS 5-7] Inject Focused Expertise:** Craft a
  ↪ concise hint that addresses the specific flaw found.
    * *Hint for Trigger 5 (Model Completeness Error):* "I've
      ↪ noticed the solution provides a fractional number of
      ↪ cars, which isn't practical. This suggests I missed an
      ↪ integer constraint in my original model. I should
      ↪ correct this by redefining the variables as integers in
      ↪ my code and re-running it."
    * *Hint for Trigger 6 (Implementation Error):* "I've
      ↪ spotted a bug. My math model for the constraint was `A
      ↪ <= B`, but in the code I wrote `A >= B`. I need to
      ↪ correct this implementation error to match my model."

\#\#\# OUTPUT STRUCTURE (MUST use these custom tags)
<action>REPLACE_AND_CONTINUE</action>
<trigger_type>[Trigger 1 | Trigger 2 | ... | Trigger
↪ 7]</trigger_type>
<analysis>[A brief explanation of why this intervention is
↪ necessary based on the detected trigger]</analysis>
<target_text>[The exact text from the original response to be
↪ replaced]</target_text>
<hint_to_insert>[Your newly crafted hint goes
↪ here]</hint_to_insert>

(OR, if no intervention is needed)

<action>NO_INTERVENTION</action>

\#\#\# --- START OF TASK ---

\#\#\# TASK_DEFINITION:
```text
{task_definition}
```

\#\#\# GROUND_TRUTH_ANSWER (if available)
The known correct final answer for the objective function is:
↪ `\\boxed{[ground_truth_answer]}`

You should use this ground truth to definitively verify the
↪ numerical correctness of the LRM's final boxed answer. If
↪ the LRM's answer is incorrect, your primary goal is to
↪ identify the root cause of the discrepancy.

```
\#\#\# LLM RESPONSE TO REFINE:
```text
{llm_response_text}
```

## J FLAW QUANTIFICATION OF NATIVE LRMS

To achieve a scalable and consistent analysis across thousands of model responses, we utilized Gemini-2.5-Pro as an expert annotator. Its task was to classify flaws in the native LRM's generated trajectories based on the seven pre-defined categories described in Appendix D.

**Distinction from the CALM Intervener.** It is crucial to distinguish this analytical use of an external model from its role as the dynamic **Intervener** within our CALM data generation framework (Section 3.2).

- **For Quantification (here):** The model acts as a **static classifier**. Its goal is to analyze a completed response and output a structured list of detected flaws for measurement purposes. It does not interact with the LRM.

- **For CALM Intervention (Section 3):** The model acts as an **interactive agent**. Its goal is to monitor a reasoning process in real-time and inject corrective hints to guide the LRM towards a better solution, thereby generating new training data.

While both roles leverage the same underlying understanding of OR modeling flaws, their functions and objectives within our study are entirely separate.

**Prompt for Flaw Classification.** The prompt below was used to guide the Gemini-2.5-Pro model in its role as a static classifier.

---

**Prompt for Flaw Classification**

```
### CONTEXT AND GOAL
You are an expert Operations Research (OR) engineer and an LLM
↪  Reasoning Pattern Analyst. Your mission is to assist in
↪  generating high-quality training data for fine-tuning Large
↪  Reasoning Models (LRMs).

The ultimate goal is to adapt an LRM's native reasoning pattern
↪  (which is heavily reliant on long-form natural language) to
↪  better emulate the iterative workflow of a human OR expert.
↪  The ideal expert workflow is a cycle of: **1. Understand &
↪  Model -> 2. Code Solver -> 3. Execute & Observe -> 4.
↪  Reflect & Debug -> (Repeat)**.

Your specific task is to analyze a given LRM response and, if it
↪  deviates from this ideal workflow, identify all the triggers
↪  we defined.

### INSTRUCTIONS
1.  First, carefully review the `TASK_DEFINITION` which contains
↪  the original problem and instructions given to the LRM.
2.  Next, analyze the provided `LLM_RESPONSE_TO_REFINE`.
3.  Identify at most two deviation based on the triggers defined
↪  below.
4.  If an deviation is found, your output MUST be structured
↪  using the custom tags `<trigger_type>`.
```

```
 5.  The only one found trigger should in <trigger_type>, for
 ↪   example, <trigger_type>Trigger 1</trigger_type>. If there
 ↪   are multiple triggers, separate them by `;`, for example,
 ↪   <trigger_type>Trigger 1;Trigger 7</trigger_type>.
 6.  If the response is ideal, your output should simply be
 ↪   <trigger_type>Correct</trigger_type>.

### DEVIATION TRIGGERS
*    **Trigger 1: Premature NL Solving:** After formulating the
 ↪   mathematical model, the LRM starts solving it manually with
 ↪   natural language instead of immediately writing solver code.
*    **Trigger 2: Fragmented Coding:** The LRM writes small,
 ↪   non-executable, or multiple solver-running code blocks
 ↪   instead of a single, comprehensive one.
*    **Trigger 3: Redundant Manual Verification:** After a code
 ↪   output, the LRM manually re-calculates the exact numerical
 ↪   results that were already provided by the solver.
*    **Trigger 4: Lack of Sanity Check/Reflection:** The LRM gets
 ↪   a correct code output but proceeds directly to the final
 ↪   answer without any high-level reflection on the result's
 ↪   plausibility.
*    **Trigger 5: Flawed Reasoning or Modeling:** The LRM's logic
 ↪   is flawed, leading to an incorrect answer. This includes
 ↪   semantic misunderstanding, a wrong mathematical model, or
 ↪   missing constraints (e.g., integers).
*    **Trigger 6: Implementation Error:** The mathematical model
 ↪   is correct, but the code is buggy or does not faithfully
 ↪   represent the model, leading to an incorrect answer.
*    **Trigger 7: Protocol Violation:** The LRM violates a clear
 ↪   instruction, especially regarding the final boxing
 ↪   requirement.

### OUTPUT STRUCTURE (MUST use these custom tags)
<trigger_type>[Trigger 1 | Trigger 2 | ... | Trigger
 ↪   7];...;[Trigger 1 | Trigger 2 | ... | Trigger
 ↪   7]</trigger_type>

### --- START OF TASK ---

### TASK_DEFINITION:
```text
{task_definition}
```
### LLM RESPONSE TO REFINE:
```text
{llm_response_text}
```
```

**Validation of the LLM Annotator.** To ensure the reliability of the automated quantification process, we validated the LLM annotator's performance against human labels. We randomly sampled 30 responses from the test set, which were independently annotated by both the LLM (using the aforementioned prompt) and one of our expert human annotators.

The agreement between the LLM and human labels was then measured. The LLM achieved an accuracy of 93.3% in identifying and correctly classifying the flaw types present in the responses, calculated based on the instance-level matching of flaw categories. This high level of agreement provides strong evidence for the validity of using the LLM for scalable and consistent flaw quantification across the entire benchmark suite.

## K  ROLE OF LARGE LANGUAGE MODELS (LLMs) IN PREPARATION OF THE MANUSCRIPT

In adherence to the ICLR 2026 submission guidelines, we hereby clarify the role of Large Language Models (LLMs) in the preparation of this manuscript.

LLMs were utilized as a general-purpose assistive tool, primarily for the purpose of **language polishing and refinement**. Specifically, we employed LLMs to improve the clarity, conciseness, and grammatical correctness of the text. The process involved providing drafted passages to an LLM and requesting suggestions for alternative phrasing, sentence restructuring, and vocabulary enhancement to better convey our intended meaning.

It is important to state that all core research ideas, experimental design, data analysis, and the primary drafting of the manuscript were conducted exclusively by the human authors. The LLM's role was strictly confined to that of a writing assistant, and it did not contribute to any of the scientific or conceptual aspects of this work.

The authors have carefully reviewed and edited all LLM-generated suggestions to ensure they accurately reflect our research and findings. We take full responsibility for all content presented in this paper, including any text that was refined with the assistance of an LLM.

