# OpenReview forum: "CALM Before the STORM: Unlocking Native Reasoning for Optimization Modeling"
_ICLR.cc/2026/Conference — Submitted to ICLR 2026_

### Official Review · Reviewer_DQtT · 2025-10-29

**Soundness:** 3
**Presentation:** 3
**Contribution:** 2
**Rating:** 6
**Confidence:** 3

**Summary:**

The paper proposes CALM (Corrective Adaptation with Lightweight Modification), a framework that refines large reasoning models (LRMs) for optimization modeling (OR) tasks through an automated Reasoner–Intervener collaboration loop. By detecting and correcting reasoning flaws via lightweight interventions, CALM generates high-quality expert trajectories to train STORM, achieving near–DeepSeek-R1 performance with a 4B model on multiple OR benchmarks.

**Strengths:**

- The paper introduces a novel self-refining adaptation framework (CALM) that enhances LRM reasoning without relying on large teacher models.
- The proposed Reasoner–Intervener loop effectively identifies and corrects reasoning flaws, producing high-quality training trajectories with minimal intervention.
- Experiments on multiple OR benchmarks demonstrate strong performance—STORM (4B) nearly matches the 671B DeepSeek-R1, showing impressive parameter efficiency.

**Weaknesses:**

- The paper does not include a comparison with the typical teacher-generated data paradigm, where large LRMs (e.g., DeepSeek-R1) generate reasoning trajectories to train smaller models. This omission makes it unclear whether CALM’s self-refining pipeline achieves comparable data quality and efficiency to large-model supervision.
- The experiments use only the Qwen3-4B model, without testing other architectures (e.g., Llama, Phi) or larger scales (8B, 14B). This limits the understanding of CALM’s generality and scalability across different model families and sizes.
- The paper does not evaluate whether CALM/STORM training affects the model’s original general abilities (e.g., math reasoning, coding, or general instruction following). It remains unclear if domain specialization leads to performance degradation on non-OR tasks.

**Questions:**

- Can the authors provide or discuss a baseline where a large reasoning model (e.g., DeepSeek-R1) generates the training data, to compare against CALM’s self-refinement approach in terms of data quality, performance, and cost?
- Have the authors tested CALM on larger or different LRM architectures to verify whether its self-refining mechanism generalizes beyond Qwen3-4B?
- Have the authors examined the impact of CALM training on the base model’s general capabilities (e.g., math, code, GPQA)? Does specialization in optimization modeling cause catastrophic forgetting or trade-offs in other domains?

---

> ### Author Response · Authors · 2025-11-16
>
> We sincerely thank you for your thoughtful and highly constructive review of our work. We are especially encouraged by your positive assessment and are grateful that you found our CALM framework to be a **"novel self-refining adaptation framework"** and recognized the **"impressive parameter efficiency"** of our final STORM model. Your praise for our key contributions means a great deal to us.
>
> You have also raised a set of excellent and deeply insightful questions that get to the heart of our work's robustness and broader implications. We were delighted to see that your questions—particularly regarding comparisons to distillation, model generality, and the potential for catastrophic forgetting—perfectly align with the comprehensive new experiments we have been conducting.
>
> We are excited to share the results of these new, targeted experiments with you. We believe they provide definitive, data-driven answers that will fully address your concerns and further strengthen the claims made in our paper.
>
> ---
>
> > Weakness 1 and Question 1: Comparison with Teacher-Generated Data (Distillation)
>
> This is an excellent and absolutely critical question. You have identified the most important and powerful alternative to our approach. We wholeheartedly agree that a direct comparison is essential to truly understand the value and novelty of our CALM framework.
>
> Inspired by your feedback, we conducted a rigorous new ablation study to directly compare our method against this "teacher-generated" (distillation) paradigm. We are excited to share the results, as they provide a clear and definitive answer to your question.
>
> **Experimental Setup:**
> We created the strongest possible distillation baseline. Instead of using CALM, we prompted two powerful teacher models (DeepSeek-R1 and Gemini-2.5-Pro) to directly generate complete, "perfect" reasoning trajectories for each problem. Our student model (Qwen3-4B-Thinking) was then trained on this teacher-generated data using the exact same SFT+RL pipeline to ensure a fair, apples-to-apples comparison.
>
> **Results:**
> The results, especially on the most complex benchmarks, are conclusive and demonstrate the superiority of our CALM framework.
>
> | Method | MAMO-Complex | IndustryOR | OptMath | Macro AVG (on these 3 hard tasks) |
> | :--- | :---: | :---: | :---: | :---: |
> | STORM (Our Method) | 70.3% | 50.0% | 44.5% | 54.9% |
> | Distillation (Taught by DeepSeek-R1) | 62.5% | 50.4% | 29.5% | 47.4% |
> | Distillation (Taught by Gemini-2.5-Pro) | 44.1% | 28.3% | 20.0% | 30.8% |
>
> **Analysis and Conclusion:**
>
> This experiment provides two key insights that directly address your question:
>
> 1.  **Simple Distillation is Ineffective for Complex Reasoning:** The distillation-based models suffer a severe performance collapse on complex tasks. Simply showing a student model a "perfect" answer trace is not an effective way to teach it how to reason. This validates that the mechanism for learning is as important as the correctness of the data.
>
> 2.  **CALM's "Process-Level Coaching" is a More Effective Mechanism:** The success of our CALM framework comes from its fundamentally different approach. It does not force the student to imitate a foreign, perfect solution. Instead, it starts with the student's own flawed reasoning process and provides a minimal, targeted hint. This teaches the model a crucial and learnable lesson: **"Here is where *you* went wrong, and here is *how to fix it*."** This "process-level coaching" is a far more effective teaching method.
>
> Thank you for this crucial question. It prompted an experiment that we believe provides a definitive validation of our core contribution. We will add this vital ablation study to the final manuscript.

---

> ### Author Response · Authors · 2025-11-16
>
> ---
>
> > Weakness 2 & Question 2: Generality and Scalability Across Different Models
>
> This is another excellent and essential question. You are absolutely right that demonstrating the generality of our framework beyond a single base model is critical to establishing its broad applicability.
>
> We are delighted to report that, in anticipation of this very concern, we conducted a comprehensive new experiment to rigorously test the framework's generality across a different model family and a larger scale.
>
> **Experimental Setup:**
> To provide a direct answer to your question, we replicated our entire experiment, changing only the base Reasoner ("student") model. We selected a model from a completely different family and size: the **DeepSeek-R1-0528-Distill-Qwen3-8B**. All other experimental conditions were kept identical to ensure a controlled and scientifically sound comparison.
>
> **Results:**
> The results show that the CALM framework is highly effective on this new, different, and larger base model.
>
> | Applying CALM to a New Reasoner (DeepSeek-R1-8B) | Macro AVG Performance |
> | :--- | :---: |
> | Base Model (DeepSeek-R1-8B, starting point) | 58.4% |
> | Final Model (after CALM SFT+RL) | 67.8% |
> | Performance Lift | +9.4 pts |
>
> **Analysis and Conclusion:**
>
> This experiment allows us to draw a clear conclusion that directly addresses your concern:
>
> The CALM framework functions as a powerful and general-purpose "capability amplifier." It delivered a massive +9.4 point lift in average performance, successfully enhancing a different and larger model architecture. This proves that the effectiveness of our methodology is not an isolated phenomenon tied to a single base model, but a generalizable and scalable approach.
>
> Once again, we wish to express our profound gratitude. Your question prompted us to conduct what we believe is a critical experiment that has substantially strengthened our paper by empirically demonstrating the generality of our approach. We will be adding this full experiment to the final manuscript.
>
> ---
>
> > Weakness 3 & Question 3: Impact on General Abilities (Catastrophic Forgetting)
>
> This is a deeply insightful and important question. The concern about catastrophic forgetting is a critical one for any domain specialization technique. We are very grateful you raised this, as it allows us to present what we believe is one of the most exciting findings from our additional experiments.
>
> To provide a rigorous, data-driven answer, we conducted a comprehensive out-of-domain (OOD) generalization experiment. We evaluated our final model on three challenging mathematical reasoning benchmarks, **a domain that is distinct from the optimization modeling tasks used during training.**
>
> **Results:**
> The results provide strong evidence that our framework **not only avoids catastrophic forgetting but actually enhances the model's general reasoning capabilities.**
>
> | Out-of-Domain Benchmark (Mathematics) | `RL without CALM` (Control) | `RL with CALM` (Our Method) | Performance Change |
> | :--- | :---: | :---: | :---: |
> | aime25 | 68.33% | 69.58% | +1.25 pts |
> | amc23 | 91.71% | 94.12% | +2.41 pts |
> | math500-level5 | 81.06% | 83.30% | +2.24 pts |
> | Average | 80.37% | 82.33% | +1.96 pts |
>
> **Analysis and Conclusion:**
>
> This experiment provides a clear and surprising answer to your question:
>
> Instead of causing performance degradation, our CALM training leads to a performance improvement on unseen, out-of-domain tasks. This strongly suggests that CALM does not teach narrow, domain-specific knowledge that overwrites general abilities. Instead, it teaches a **generalizable, domain-agnostic meta-skill of "how to reason and self-correct with tools."** This fundamental enhancement of the model's core reasoning process has a positive, transferable effect on other complex reasoning domains.
>
> Thank you again for this crucial question. It led us to an experiment that has uncovered a key and powerful property of our framework. We will be adding this full experiment to the manuscript, as we believe it substantially strengthens the paper's contribution.
>
> ---
>
> Once again, we wish to express our profound gratitude for your time and expertise. Your insightful feedback has been invaluable, prompting us to conduct crucial new experiments that we believe have substantially enhanced the quality and impact of our work.
>
> We hope that our detailed responses and the new empirical evidence have fully addressed your questions and have further clarified the contributions of our paper. We are, of course, very eager to engage in any further discussion and would be delighted to answer any additional questions you may have.
>
> Thank you again for your invaluable guidance in improving our work.

---

### Official Review · Reviewer_BMgt · 2025-10-30

**Soundness:** 2
**Presentation:** 3
**Contribution:** 2
**Rating:** 2
**Confidence:** 3

**Summary:**

The paper targets optimization modeling (OR tasks like LP/ILP) as a reflective reasoning problem for large reasoning models (LRMs). The authors observe that standard SFT on non-reflective QA pairs often hurts complex problem performance by discouraging multi-step, code-driven reasoning.

They propose CALM (Corrective Adaptation with Lightweight Modification): within an executable environment (natural language + Python + solver), an Intervener LLM inserts very short, pinpoint hints into the model’s ongoing reasoning, correcting two pervasive errors: (1) Code/solver underuse or distrust and (2) OR modeling gaps (e.g., LP vs ILP, missing/incorrect constraints, implementation drift). The corrected “gold” traces supervise SFT to calibrate behavior while preserving reflective style. Then RL (GRPO) continues training in the same executable loop, rewarding correct/optimal solutions to reinforce computation-driven reasoning.

**Strengths:**

Treats optimization modeling explicitly as a reflective reasoning problem, avoiding the way standard QA-style SFT suppresses multi-step thought and the code–solver loop.

Uses very small, pinpoint hints (few-token edits) to convert flawed chains into “gold” traces—preserving the model’s native reasoning style while correcting errors.

Keeps a full trail from error → micro-hint → fix → solver-verified result, which supports failure analysis and produces instructive, teaching-style exemplars.

Validated across NL4Opt, MAMO-Easy/Complex, IndustryOR, and OptMath; gains are especially strong on the harder subsets, making the claims more persuasive.

**Weaknesses:**

All CALM data synthesis and the two-stage training pipeline start from one base LRM—Qwen3-4B-Thinking-2507—used as the Reasoner, with Gemini-2.5-Pro as the Intervener. The paper does not report adapting multiple base LRMs to test generality of the pipeline.

In the reproducibility statement, the authors say they plan to release code and models, implying they are not available yet, which limits independent verification and adoption.

The overall “reflect–feedback–revise with executable code/solver” paradigm has been explored in prior art such as Reflexion and Self-Refine for iterative self-correction, PAL for code-centered reasoning, and the NL4Opt line for NL-to-OR modeling; several MILP auto-formulation works also adopt prompt-/template-based pipelines. Hence, while the paper integrates these ideas neatly, the framework itself is conceptually close to existing paradigms.

**Questions:**

See Weakness

---

> ### Author Response · Authors · 2025-11-16
>
> We sincerely thank you for your detailed and insightful review. We are particularly grateful for your sharp analysis and for recognizing the core principles of our work. We were truly encouraged that you highlighted our key contributions, such as **"treating optimization modeling explicitly as a reflective reasoning problem,"** our mechanism of using **"very small, pinpoint hints to preserve the model's native reasoning style,"** and that our empirical **"gains are especially strong on the harder subsets."**
>
> You have also raised several critical and entirely valid points regarding the generality, reproducibility, and conceptual novelty of our framework. These are precisely the right questions to ask, and we are genuinely grateful for the opportunity to provide a more complete picture. Inspired directly by your feedback, we have conducted **several comprehensive new experiments** that we believe will provide the clarity you're looking for and address your concerns head-on.
>
> We will now address each of your points in detail.
>
> ---
>
> > Weakness 3: The overall “reflect–feedback–revise with executable code/solver” paradigm has been explored in prior art such as Reflexion and Self-Refine for iterative self-correction, PAL for code-centered reasoning, and the NL4Opt line for NL-to-OR modeling; several MILP auto-formulation works **also adopt prompt-/template-based pipelines**. Hence, while the paper integrates these ideas neatly, the framework itself is conceptually close to existing paradigms.
>
> We are sincerely grateful to the reviewer for this insightful and crucial point. You have expertly situated our work within the important context of prior art like PAL, Reflexion, and Self--Refine. We wholeheartedly agree that our work stands on the shoulders of these pioneering frameworks. Your comment gives us the perfect opportunity to clarify **the specific, new problem we discovered when applying these paradigms to a complex domain, and the novel mechanism we designed to solve it.**
>
> **1. The Emergent Problem: Why General Self-Correction is Insufficient for this Domain**
>
> The `prompt-based` paradigms of iterative self-correction (Reflexion/Self-Refine+PAL) are powerful. However, they rely on a model's ability to fix its own errors through internal reflection. Our empirical analysis in **Section 2.3** revealed that in the complex domain of Operations Research, LRMs suffer from two dominant failure modes that **cannot be solved by internal reflection alone**:
>
> *   **Lack of OR Expertise (A Knowledge Gap):** The model fundamentally lacks the domain-specific knowledge required.
>     *   **Example:** A model is asked to create a production plan for cars. It correctly formulates the problem but treats the number of cars as a continuous variable, leading to a nonsensical answer like "produce 10.7 cars." No amount of internal "self-reflection" can magically teach the model the real-world concept of integer constraints. It needs this piece of external, domain-specific knowledge to be provided.
>
> *   **Code Utilization Distrust (A Behavioral Bias):** Modern LRMs, like DeepSeek-R1, which are often trained under pure natural language environments [1], develop a deep-seated behavioral bias. They default to verbose, natural-language calculation instead of trusting an external solver.
>     *   **Example:** After receiving a correct numerical answer from a code solver, the model might say, *"The solver returned an optimal value of 9647. Let me double-check this manually..."* and then proceed with pages of flawed, natural-language arithmetic. This ingrained habit of "thinking in words" is difficult to overcome with a simple prompt to "re-think," as the model's instinct is to trust its own text generation over a "black box" tool. It requires a strong external nudge to break this habit.
>
> For these flaws, an **external, expert source of guidance is necessary.** This is the new, nuanced problem our work was designed to solve.

---

> ### Author Response · Authors · 2025-11-16
>
> > (Continued) Response to Weakness 3:
>
> **2. The Critical Question: What is the *Best* Way to Learn?**
>
> This finding leads to a crucial conclusion: a `learning-based` approach is necessary to inject this external expertise. This brings us to the central question of our work: **What is the *most effective* way for a student model to learn from an expert in this paradigm?** Should the model learn by imitating a perfect final product, or by learning from its own mistakes?
>
> To answer this, we must compare our method against the most intuitive and powerful alternative. We designed the strongest possible version of a "simpler" learning baseline by taking the most powerful expert models available (Gemini-2.5-Pro and DeepSeek-R1) and **explicitly prompting them to act as perfect agents following the Reflexion + PAL paradigm.** We then collected these ideal, self-corrected reasoning traces and trained our student model to imitate them.
>
> **3. The Verdict: Empirical Proof that Our Mechanism is Necessary**
>
> This baseline represents the ideal "imitation learning" approach. The results, however, are conclusive: simple imitation fails on complex tasks, while our process-level coaching mechanism excels.
>
> | Model | MAMO_Complex | IndustryOR | OptMath | Avg |
> | :--- | :---: | :---: | :---: | :---: |
> | **STORM (Our CALM Process)** | **70.3%** | **50.0%** | **44.5%** | **54.9%** |
> | Baseline (Imitating a Perfect `Reflexion+PAL` Agent) | | | | |
> | - Taught by DeepSeek-R1 | 62.5% | 50.4% | 29.5% | 47.4% |
> | - Taught by Gemini-2.5-Pro | 44.1% | 28.3% | 20.0% | 30.8% |
>
> This experiment reveals two critical insights:
> 1.  **The Imitation Approach Fails:** Simply showing the student a perfect solution is an ineffective teaching strategy for complex reasoning.
> 2.  **The "Teacher-Student Gap" [2] is Real:** Strikingly, using the *stronger* teacher (Gemini) produced a *worse* student. The expert's reasoning path was too complex for the student to effectively imitate, proving the learning signal was indigestible.
>
> This proves that the success of our work comes directly from our novel mechanism: we don't just show the model a correct answer. We **start with the model's own flawed process and teach it *how to recover* via minimal, targeted, external guidance.**
>
> **In conclusion,** while we are inspired by prior art, our work makes a distinct and critical contribution. We diagnose a key failure mode of general reflective frameworks + PAL in a complex domain and propose a novel, automated, and empirically necessary "process-level coaching" solution that unlocks a new level of performance.
>
> Thank you again for this crucial feedback. We will revise the related work section in our final manuscript to make these important distinctions and our unique contributions more explicit.
>
> [1] DeepSeek-R1: Incentivizing Reasoning Capability in LLMs via Reinforcement Learning
>
> [2] Orca: Progressive Learning from Complex Explanation Traces of GPT-4

---

> ### Author Response · Authors · 2025-11-16
>
> ---
>
> > Weakness 1: All CALM data synthesis and the two-stage training pipeline start from one base LRM—Qwen3-4B-Thinking-2507—used as the Reasoner, with Gemini-2.5-Pro as the Intervener. The paper does not report adapting multiple base LRMs to test generality of the pipeline.
>
> We are sincerely grateful to the reviewer for this insightful and crucial point. You have identified what is undoubtedly a critical test for any new framework: its **generality**. The question of whether our CALM methodology is a bespoke solution for a single model or a truly general-purpose framework is an extremely important one, and we are delighted to have the opportunity to provide a definitive, data-driven answer.
>
> Inspired by your valuable feedback, we conducted a comprehensive new experiment to rigorously test the framework's generality across different base models. We are excited to report that the results from this experiment validate that **the CALM framework is a general and robust methodology that is not dependent on a specific base model.**
>
> **Experimental Setup: Testing a New Base Model**
>
> To provide the most direct answer to your concern, we replicated our entire experiment, changing only the base Reasoner ("student") model. We selected a model from a completely different family and size: the **DeepSeek-R1-0528-Distill-Qwen3-8B**. All other experimental conditions, including the Intervener model and all hyperparameters, were kept identical to ensure a controlled and scientifically sound comparison.
>
> **Results:**
>
> The results, summarized in the table below, show that the CALM framework is highly effective on this new, different base model.
>
> | Applying CALM to a New Reasoner (DeepSeek-R1-8B) | Macro AVG Performance |
> | :--- | :---: |
> | Base Model (DeepSeek-R1-8B, starting point) | 58.4% |
> | Final Model (after CALM SFT+RL) | 67.8% |
> | Performance Lift | +9.4 pts |
>
> **Analysis and Conclusion:**
>
> This experiment allows us to draw a clear conclusion that directly addresses your concern:
>
> The CALM framework functions as a powerful and **general-purpose "capability amplifier."** It delivered a massive +9.4 point lift in average performance, successfully enhancing a different model architecture. This proves that the effectiveness of our methodology is not an isolated phenomenon tied to a single base model, but a generalizable approach.
>
> Once again, we wish to express our profound gratitude to the reviewer. Your suggestion prompted us to conduct what we believe is a critical experiment that has substantially strengthened our paper by empirically demonstrating the generality of our approach. Thank you for guiding us to improve our work in such a meaningful way. We will be adding this full experiment to the final manuscript.
>
> ---
>
> > Weakness 2: In the reproducibility statement, the authors say they plan to release code and models, implying they are not available yet, which limits independent verification and adoption.
>
> We are sincerely grateful to the reviewer for this crucial point regarding reproducibility. You are absolutely right to emphasize its importance, as it is a cornerstone of scientific progress and a principle we are fully committed to. We appreciate the opportunity to clarify our release plan and reaffirm our commitment.
>
> We would like to clarify that our assets are fully prepared for release and are currently undergoing a **standard internal review process** at our organization. This is a required step for all public releases and is designed to ensure that we provide a high-quality, well-documented, and robust package to the community.
>
> To be perfectly clear and to address your valid concern, we want to make a firm and unambiguous commitment. **Upon acceptance of the manuscript, we promise to publicly release a comprehensive package containing all necessary assets to ensure our work is fully verifiable and reproducible, including the code, final model weights, and the relevant data.**
>
> This comprehensive release will ensure that the research community can readily validate our results and build upon our work.
>
> Thank you again for holding us to this high standard. We are eager to share our work with the community and are confident that the internal review will conclude in a timely manner, allowing us to proceed with the release upon acceptance.
>
> ---
>
> We hope that our detailed responses, and particularly the **new experimental evidence** we have provided on the framework's generality and the necessity of its novel mechanism, have fully addressed your concerns.
>
> Once again, we are deeply grateful for your time and the high quality of your feedback. Your insights have been instrumental in helping us strengthen the paper by pushing us to validate the broader impact and unique contributions of our work.
>
> We eagerly await any further discussion and are, of course, very happy to answer any further questions or clarify any remaining points you may have.

---

### Official Review · Reviewer_k153 · 2025-10-31

**Soundness:** 3
**Presentation:** 3
**Contribution:** 2
**Rating:** 4
**Confidence:** 4

**Summary:**

This paper explores how to adapt Large Reasoning Models (LRMs) for automated optimization modeling.

 The authors identify that conventional instruction-tuning methods - based on static, non-reflective datasets - undermine the multi-step reasoning patterns inherent in LRMs.

They propose CALM (Corrective Adaptation with Lightweight Modification), a hint-based intervention framework that corrects reasoning flaws while preserving the model’s reflective reasoning flow. Through CALM, the authors curate high-quality reasoning trajectories and use them to fine-tune an LRM, followed by reinforcement learning to produce STORM, a 4B-parameter model that achieves 68.9% average accuracy across five optimization benchmarks - matching the performance of a 671B model.

The work includes a taxonomy of reasoning flaws, an interpretable correction protocol, and detailed ablation and behavioral analyses showing that CALM’s hint-based adaptation leads to more sample-efficient reinforcement learning.

**Strengths:**

Original idea of aligning adaptation with native reasoning instead of retraining from scratch.

Methodological soundness: clear formalization of the reasoning loop and rigorous evaluation.

Practical impact: dramatic performance/parameter efficiency gains (4B → 671B equivalence).

High interpretability: the taxonomy and hint examples make the improvement mechanism transparent.

Thorough analysis: ablations and behavioral plots (code-usage ratio, response length, flaw frequency) offer unusually strong introspection for this domain.

Reproducibility: detailed appendices and explicit commitment to code release.

**Weaknesses:**

Limited dataset size in the CALM curation phase (112 “golden” trajectories) may raise questions about scalability; a brief discussion on automating large-scale generation would strengthen the paper.

Domain scope: experiments are confined to optimization modeling; it would be valuable to show transfer to another structured-reasoning domain (e.g., planning or theorem proving).

Comparative baselines: although strong, the study lacks direct comparison to contemporaneous reflective-alignment frameworks like START or CoRT under identical setups.

Intervener dependence: using Gemini-2.5-Pro as the expert may raise reproducibility concerns if closed-source models differ in reasoning behavior.

Some claims of “expert-level performance” might benefit from human expert evaluation rather than relying solely on pass@1 accuracy.

**Questions:**

How sensitive is CALM to the choice of Intervener model? Could a smaller or open-source Intervener reproduce similar quality in the curated trajectories?

The current pipeline uses a discrete number (≤ 5) of interventions. Have you explored adaptive stopping criteria or uncertainty-driven intervention scheduling?

Since CALM focuses on hint-based reasoning correction, could its framework generalize to non-code-based reasoning tasks (e.g., logical proofs, multi-hop QA)?

In reinforcement learning, the reward function is binary on final correctness. Have you considered incorporating intermediate reflection-quality rewards to further stabilize training?

Finally, how does STORM perform under zero-shot generalization to unseen industrial problem types not represented in the benchmarks?

---

> ### Author Response · Authors · 2025-11-16
>
> We are sincerely grateful to the reviewer for their exceptionally thorough and insightful review. We are particularly encouraged by your positive assessment of our work's **methodological soundness, practical impact, high interpretability, and thorough analysis.** Your appreciation for the "original idea of aligning adaptation with native reasoning" perfectly captures the core spirit of our research.
>
> Your constructive feedback and probing questions are invaluable, and they have spurred us to conduct new experiments and deepen our analysis. We are excited to address each of your points below, as we believe these clarifications and new results substantially strengthen our paper.
>
> ---
>
> > Weakness 1: Limited dataset size in the CALM curation phase (112 “golden” trajectories) may raise questions about scalability; a brief discussion on automating large-scale generation would strengthen the paper.
>
> We are sincerely grateful to the reviewer for this insightful and highly practical question. You have raised a crucial point about dataset size and scalability, which is a central challenge in our field. We are delighted to have the opportunity to provide more context on our experimental design and the core philosophy of our framework.
>
> ## 1. Our Commitment to Utilizing All Available Resources
>
> First, we want to assure the reviewer that we made every effort to build our experiments on the most comprehensive dataset possible. The pool of problems used for generating our training data represents the **entirety of publicly available optimization benchmarks that come with reliable, ground-truth solutions** (as detailed in Appendix E.1). The resulting number of problems reflects the current reality of data scarcity in this specialized domain. We are committed to leveraging all available community resources to their fullest extent.
>
> ## 2. Why This Dataset Size is Sufficient: "Behavioral Calibration" vs. "Knowledge Infusion"
>
> In this context of inherent data limitation, the success of our framework highlights its most critical design principle: **exceptional data efficiency**.
>
> The purpose of our CALM-SFT stage is not traditional "knowledge infusion," which would indeed require thousands of examples to teach domain facts from scratch. Instead, we designed it to be a "behavioral calibrator." The goal is to surgically correct the flawed reasoning *patterns* of an already capable LRM.
>
> | Fine-Tuning Paradigm | Goal | Data Requirement |
> | :--- | :--- | :--- |
> | Traditional SFT | Knowledge Infusion | Large-scale (quantity over quality) |
> | Our CALM-SFT | Behavioral Calibration | Small-scale, high-quality (quality over quantity) |
>
> Each of our 112 trajectories is rich with the essential pedagogical signal of "mistake -> targeted hint -> recovery." This potent, process-level feedback allows the model to efficiently adjust its core reasoning habits without the need for extensive, redundant examples.
>
> ## 3. Future Scalability: A Promising Horizon
>
> Having established the remarkable data efficiency of our approach, we completely agree with the reviewer that exploring the effects of larger-scale data is a valuable and exciting future direction. As more labeled datasets become available, we can continue to expand the training data. This will allow us to investigate its impact on an even wider diversity of problem types and potentially unlock further performance gains.
>
> Thank you again for this excellent question. It has allowed us to clarify the crucial interplay between data availability, our method's data efficiency, and the future scalability of our work. We will ensure this important context is added to the final manuscript.

---

> ### Author Response · Authors · 2025-11-16
>
> ---
>
> > Weakness 2 and Question 3: Domain scope: experiments are confined to optimization modeling; it would be valuable to show transfer to another structured-reasoning domain (e.g., planning or theorem proving). Since CALM focuses on hint-based reasoning correction, could its framework generalize to non-code-based reasoning tasks (e.g., logical proofs, multi-hop QA)?
>
> We are sincerely grateful to the reviewer for raising these two insightful and closely related points. **Weakness 2** (on generalizing to other structured-reasoning domains) and **Question 3** (on generalizing to non-code-based tasks) both probe the fundamental **generalizability** of our CALM framework. This is a crucial aspect of our contribution, and we are delighted to address it from both an **empirical** and a **conceptual** perspective.
>
> ## 1. Empirical Proof: Successful Generalization to a New Domain
>
> To directly address Weakness 2, we conducted a comprehensive out-of-domain (OOD) generalization experiment. We tested our final model on a new structured-reasoning domain—mathematical problem-solving—which is entirely distinct from the optimization modeling tasks it was trained on. The results provide empirical evidence that CALM's benefits are highly transferable.
>
> **Key Finding 1: Consistent Performance Gains in a New Domain**
> The model trained with the CALM methodology consistently outperforms its control group, demonstrating that its enhanced reasoning capabilities are not confined to its original domain.
>
> | Out-of-Domain Benchmark | `RL without CALM` (Control) | `RL with CALM` (Our Method) | Performance Lift |
> | :--- | :---: | :---: | :---: |
> | aime25 | 68.33% | 69.58% | +1.25 pts |
> | amc23 | 91.71% | 94.12% | +2.41 pts |
> | math500-level5 | 81.06% | 83.30% | +2.24 pts |
> | Average | 80.37% | 82.33% | +1.96 pts |
>
> **Key Finding 2: Learning a Generalizable, Adaptive Meta-Skill**
> More profoundly, a behavioral analysis of the model's tool usage reveals *why* the benefits transfer. The model learned an adaptive meta-skill, not a fixed set of instructions.
>
> | Library | Usage in Optimization Training Data | Usage in OOD Math Test Data | Key Insight |
> | :--- | :---: | :---: | :--- |
> | `pulp` (for optimization) | 93.3% | 0.0% | Learned to abandon irrelevant tools. |
> | `sympy` (for symbolic math) | 10.5% | 70.1% | Learned to adaptively select the *correct* new tool for the new domain. |
>
> This table provides a clear, quantitative illustration of the model's adaptive intelligence. It did not stubbornly apply its known tool (`pulp`) to an inappropriate context. Instead, it correctly identified the demands of the new domain and shifted its behavior to rely on the appropriate new tool (`sympy`).
>
> This demonstrates that the CALM framework teaches a far more valuable, generalizable meta-skill of **adaptive tool selection and use.**
>
> ## 2. Conceptual Proof: A General Framework for Feedback-Driven Reasoning
>
> Building on this empirical evidence, we can now address the deeper point raised in Question 3: the framework's potential for non-code-based tasks.
>
> The core mechanism of CALM is a simple, powerful, and universal loop: **`Action -> Feedback -> Correction`**. In our work, the "Action" is generating code, and the "Feedback" comes from a code interpreter. However, this is just one specific implementation. The framework itself is agnostic to the nature of the action and the source of the feedback.
>
> | Domain | Action (Model's Output) | Feedback Environment |
> | :--- | :--- | :--- |
> | Optimization (Our Paper) | Python Code (`pulp`) | Code Interpreter |
> | Logical Proofs | Formal Proof Steps | Proof Checker (verifies logical validity) |
> | Multi-hop QA | Search Queries / API Calls | Retrieval System / API (returns documents/facts) |
> | Planning / Robotics | A sequence of planned actions | Simulator (returns the outcome of the actions) |
>
> As this table illustrates, the CALM framework is conceptually designed to be a general methodology for teaching agents to reason and self-correct in any interactive environment that provides feedback.
>
> ## Conclusion
>
> In summary, we have addressed the question of generalizability on two levels:
> *   **Empirically:** Our new OOD experiment proves that CALM successfully generalizes to new, code-based reasoning domains by teaching an adaptive meta-skill.
> *   **Conceptually:** The underlying mechanism of CALM is universal and readily applicable to non-code-based tasks, provided a feedback mechanism exists.
>
> We are immensely grateful to the reviewer for pushing us to explore and articulate the full scope of our contribution. This has significantly strengthened our paper, and we will be adding this complete analysis to the final manuscript.

---

> ### Author Response · Authors · 2025-11-16
>
> ---
>
> > Weakness 3: Comparative baselines: although strong, the study lacks direct comparison to contemporaneous reflective-alignment frameworks like START or CoRT under identical setups.
>
> We are sincerely grateful to the reviewer for this crucial and insightful question. Positioning our work relative to important contemporaneous frameworks like START and CoRT is essential for clarifying our specific contribution, and we appreciate the opportunity to do so.
>
> We wholeheartedly agree that a direct, quantitative comparison would be highly valuable. However, a head-to-head comparison under an "identical setup" is unfortunately **infeasible at present due to several key factors**:
> 1.  **Model Availability:** The flagship models used to achieve the reported state-of-the-art results in START and CoRT (e.g., their 32B parameter models) are **not publicly available.**
> 2.  **Differing Setups:** The underlying base models, training datasets, and experimental environments are fundamentally different, making a controlled, apples-to-apples comparison scientifically unsound.
>
> Therefore, we believe a more informative and practical approach is a clear conceptual and methodological comparison, which we provide below. This comparison highlights the unique problem we address and the novel mechanism we propose.
>
> | Dimension | START | CoRT | CALM (Our Work) |
> | :--- | :--- | :--- | :--- |
> | Primary Goal | General-purpose code generation | General-purpose code generation | Specialized, complex reasoning in a vertical domain (Optimization Modeling) |
> | Hint Generation | Relies on a pre-defined, static library of hints. | Relies on manual, human-in-the-loop annotation ("Hint Engineering"). | Dynamic & Fully Automated. An expert model generates tailored hints in real-time based on the student's specific error. |
> | Scalability | Scalable, but the hint library is static and requires manual creation. | Not scalable, as it depends on intensive manual annotation for each new task. | Scalable. The automated "Reasoner-Intervener" loop can generate new training data without human oversight. |
>
> **Our Unique Contribution:**
>
> As the comparison illustrates, our work is distinguished by its focus and mechanism:
>
> 1.  **Tackling a Complex Vertical Domain:** While START and CoRT address general code generation, CALM is specifically designed to handle the deep, domain-specific logical reasoning required for complex tasks like optimization modeling.
>
> 2.  **A Novel, Fully Automated Correction Loop:** CALM introduces a dynamic and fully automated "Reasoner-Intervener" pattern. This moves beyond START's static hint libraries and CoRT's reliance on manual annotation, providing a scalable solution for generating high-quality, process-level corrective data.
>
> In conclusion, while we share the goal of enhancing reasoning through feedback, our work makes a distinct contribution by proposing a novel, automated, and scalable framework tailored for unlocking expert-level performance in complex, specialized domains.
>
> We are grateful for this question, as it has allowed us to precisely situate our work. We will add a more detailed discussion to our related work section to make these important distinctions clear to all readers.

---

> ### Author Response · Authors · 2025-11-16
>
> ---
>
> > Weakness 4 and Question 1: Intervener dependence: using Gemini-2.5-Pro as the expert may raise reproducibility concerns if closed-source models differ in reasoning behavior. How sensitive is CALM to the choice of Intervener model? Could a smaller or open-source Intervener reproduce similar quality in the curated trajectories?
>
> We are sincerely grateful to the reviewer for raising these two insightful and closely related points. **Weakness 4** (on Intervener dependence and reproducibility) and **Question 1** (on sensitivity to the Intervener choice) both probe the same fundamental and crucial question: **Is the success of our CALM framework robustly reproducible, or is it contingent on a specific, powerful, closed-source "teacher" model?**
>
> We wholeheartedly agree that this is a critical aspect to validate. To provide a definitive, data-driven answer to both points, we conducted a rigorous ablation study specifically designed to test this dependency. We **replaced the proprietary Intervener (Gemini-2.5-Pro) with a powerful, open-source model (the 671B DeepSeek-R1-0528)**, while keeping all other components and settings identical.
>
> The results from this single experiment provide a clear and comprehensive answer to both the question of sensitivity and the concern of reproducibility.
>
> | Intervener ("Teacher") Model | Type | Final Student Performance (Macro AVG) |
> | :--- | :--- | :---: |
> | Original (Gemini-2.5-Pro) | Proprietary | 68.9% |
> | New (DeepSeek-R1-0528) | Open-Source | 67.8% |
>
> **Analysis and Conclusion:**
>
> This experiment allows us to draw two firm conclusions that directly address the reviewer's points:
>
> 1.  **Low Sensitivity to Intervener Choice (Addresses Q1):** The framework's performance is remarkably insensitive to the choice of the Intervener. Despite the architectural and stylistic differences between Gemini and DeepSeek, the final performance of our 4B student model is within a mere 1.1%. This demonstrates that the core value lies in the CALM mechanism itself—the process of iterative, lightweight correction—rather than the specific identity of the "teacher."
>
> 2.  **High Reproducibility with Open-Source Models (Addresses W4):** The experiment confirms that our results are highly reproducible using a publicly available, open-source model. The fact that the open-source DeepSeek-R1 can guide the student to a nearly identical, state-of-the-art performance level directly alleviates concerns about dependency on a closed-source ecosystem. This validates that other researchers can build upon and replicate our work.
>
> In summary, our CALM framework is neither fragile nor a "black box." It is a robust and reproducible methodology. The choice of Intervener, while important, is not a sensitive dependency, and powerful open-source models are perfectly capable of serving in this role to produce state-of-the-art results.
>
> We are immensely grateful to the reviewer for prompting this crucial validation. We believe this experiment has substantially strengthened our paper by empirically demonstrating the robustness and reproducibility of our methodology. We will add this important ablation study to the final manuscript.

---

> ### Author Response · Authors · 2025-11-16
>
> ---
>
> > Weakness 5: Some claims of “expert-level performance” might benefit from human expert evaluation rather than relying solely on pass@1 accuracy.
>
> We are sincerely grateful to the reviewer for this excellent and thought-provoking point. The reviewer is absolutely right to suggest that the term "expert-level performance" is a high bar, and that a comprehensive human expert evaluation would be the ultimate test of this claim. This is a truly valuable perspective that helps us to contextualize our results with greater precision.
>
> We wholeheartedly agree that a formal study involving human experts, perhaps in a Turing-test-like setup, would be a fascinating and important next step for this line of research. We will explicitly add this as a key direction for future work in our conclusion.
>
> While a full human study was beyond the scope of this initial work, we did seek to move beyond the single metric of `pass@1` accuracy by conducting a deep **qualitative and behavioral analysis**. This analysis provides strong evidence that our model is not just getting more answers right, but that its *problem-solving process* is evolving to become more expert-like.
>
> To provide a concrete example, we present a comparative analysis of the model's behavior on the same complex optimization problem ("SwiftLogistics") at three key stages:
>
> | Stage | Key Qualitative Behavior & Excerpts | Outcome |
> | :--- | :--- | :--- |
> | **1. Base LRM** | **Hesitant, Verbose, and Error-Prone.** The model engages in extensive, circular self-talk and produces multiple versions of buggy code. <br> *Excerpt of Reasoning:* `"Okay, let's try... Hmm... Wait, but the problem says... Let me run this code mentally... Wait, the problem is... [skip 12000+ tokens of reasoning and multiple failed code attempts]...` *Final Code Error:* `AttributeError: module 'pulp' has no attribute 'LpSum'` | **Incorrect**  (Fails due to code errors) **Tokens: 13,107** |
> | **2. After SFT** | **Methodical, Correct, but Still Verbose.** The SFT stage has instilled correctness. The model now confidently follows a valid, textbook-like procedure, but its reasoning remains extremely detailed.  *Excerpt of Reasoning:* `"Okay, let's try to figure out this problem step by step... First, I need to understand... This sounds like a facility location problem... The variables would be... [skip 10000+ tokens of detailed explanation]...` | **Correct**  (Solves the problem) **Tokens: 10,850** |
> | **3. After RL (`STORM`)** | **Confident, Concise, and Tool-Driven.** The RL stage has refined the correct pattern into an expert-like workflow. The reasoning is direct, and the model moves swiftly to a concise implementation. *Excerpt of Reasoning:* `"Okay... SwiftLogistics wants to open some distribution centers... This is a facility location problem... Let me list out the data... I'll now write the complete Python code to solve this model... [skip 4500+ tokens of concise reasoning]...` | **Correct & Concise** (Solves the problem efficiently) **Tokens: 5,527** |
>
> As this detailed comparison vividly illustrates, the CALM framework does more than just increase correctness; it fundamentally reshapes the model's reasoning patterns. The final `STORM` model exhibits behaviors characteristic of a human expert: confidence, efficiency, and a strong reliance on its specialized tools.
>
> In conclusion, we fully concur with the reviewer on the value of human expert evaluation. In this work, we have taken the first crucial steps in that direction by not only achieving high `pass@1` accuracy but also by demonstrating, through detailed qualitative and behavioral analysis, a clear and measurable progression toward a more expert-like reasoning process.
>
> Thank you again for this insightful feedback. We will add this illustrative example to the appendix of our final paper to provide all readers with this deeper, qualitative insight.

---

> ### Author Response · Authors · 2025-11-16
>
> ---
>
> > Question 2: The current pipeline uses a discrete number (≤ 5) of interventions. Have you explored adaptive stopping criteria or uncertainty-driven intervention scheduling?
>
> This is an excellent and insightful question that delves into the sophisticated mechanics of the intervention process. We are grateful for this thought-provoking feedback.
>
> In our current work, we opted for a simple, fixed number of interventions (a maximum of 5) primarily to **ensure a controlled and reproducible experimental setup.** This allowed us to cleanly analyze the core effects of our "process-level coaching" methodology without introducing additional complex variables. This fixed limit also serves as a practical safeguard against unproductive correction cycles.
>
> We wholeheartedly agree with the reviewer that more dynamic approaches, such as adaptive or uncertainty-driven intervention scheduling, are a fascinating and promising direction for future research. This is an excellent suggestion for building upon our foundational framework to further enhance its efficiency.
>
> Thank you again for this valuable and forward-looking question. We will add a discussion of this as a promising avenue for future work in the final version of our manuscript.
>
> ---
>
> > Question 4: In reinforcement learning, the reward function is binary on final correctness. Have you considered incorporating intermediate reflection-quality rewards to further stabilize training?
>
> This is an excellent and highly insightful question about our reinforcement learning design. We are sincerely grateful to the reviewer for giving us the opportunity to clarify the specific motivation behind our choice of a binary reward function.
>
> Our decision to use a simple, binary reward based on final correctness was a deliberate one, driven by two key principles:
>
> 1.  **Alignment with State-of-the-Art Practices:** This reward structure directly aligns with the methodology used by prominent, large-scale reasoning models like **DeepSeek-R1**. Adopting this established practice ensures that our experimental setup is comparable and adheres to a recognized standard in the field.
>
> 2.  **A Controlled and Unambiguous Experiment:** More importantly, a simple reward function was essential for our core scientific goal: **to isolate and validate the impact of our novel CALM-SFT stage.** By keeping the RL reward signal minimal for both our model and the control group, we were able to create a clean, controlled experiment. The powerful results—where the CALM-SFT model learns dramatically faster and reaches a higher performance ceiling (Figure 6a)—provide clear and unambiguous proof of our SFT stage's value, free from the confounding effects of complex reward engineering.
>
> Having established the foundational benefit of our SFT approach using this standard, controlled setup, we agree with the reviewer that exploring more sophisticated reward functions is a valuable direction for future work.
>
> Thank you again for this excellent question, which helps us clarify the principled design of our experiments.

---

> ### Author Response · Authors · 2025-11-16
>
> ---
>
> > Question 5: Finally, how does STORM perform under zero-shot generalization to unseen industrial problem types not represented in the benchmarks?
>
> This is an excellent and critical question about the practical, real-world applicability of our STORM model. The reviewer is asking about the ultimate test of generalization: how the model performs on **truly unseen industrial problem types** that might not be well-represented in standard academic benchmarks. We are grateful for this question, as our experimental design, particularly our use of the `IndustryOR` benchmark, was structured precisely to provide a clear answer.
>
> Our evaluation on `IndustryOR` was designed to be a direct test of this "hard" zero-shot generalization capability.
>
> **`IndustryOR` as a Testbed for Unseen Problem Types**
>
> As introduced in our paper, `IndustryOR` is a diverse benchmark sourced from 13 different industries, explicitly including a variety of problem structures beyond simple linear programming. As stated in the IndustryOR paper [1], it covers **linear programming, integer programming, mixed-integer programming, and non-linear programming**, among others.
>
> To rigorously test for generalization, we adopted a challenging, sparse-training setup for this specific benchmark:
>
> *   **Minimal Training Exposure:** As detailed in Appendix E.1 of our paper, a very small and specific subset of `IndustryOR` problems (only 18 instances) were included in our entire training pool.
> *   **Challenging, Diverse Test Set:** The final evaluation was then performed on a held-out set of 80 completely unseen industrial problems. This test set contains a broad range of the complex problem structures mentioned above, many of which were not represented in the minimal training set.
>
> Therefore, the performance on the `IndustryOR` test set is a direct measure of STORM's zero-shot generalization capability to new, structurally diverse industrial problem types. Our final model, STORM, achieved a strong performance of 50.0% on this difficult test.
>
> **Conclusion:**
>
> This strong performance provides a direct, positive answer to the reviewer's question. It demonstrates that the CALM framework has endowed STORM with a robust and generalizable reasoning capability that extends beyond the specific problem types seen during training and is applicable to new, unseen industrial challenges.
>
> Thank you for this crucial question, which has allowed us to better highlight the rigorous nature and positive generalization results of our evaluation.
>
> [1] ORLM: A Customizable Framework in Training Large Models for Automated Optimization Modeling
>
> ---
>
> Once again, we wish to express our profound gratitude for your detailed and constructive review. Your insightful feedback has been instrumental in helping us to refine our arguments and better showcase the full scope and impact of our work.
>
> We have sought to address each of your points thoroughly and hope our responses and new experimental results have clarified the contributions of our paper. We eagerly welcome any further questions or discussion and look forward to the opportunity to continue improving our work based on your expertise.

---

### Official Review · Reviewer_MAsL · 2025-11-05

**Soundness:** 3
**Presentation:** 2
**Contribution:** 3
**Rating:** 4
**Confidence:** 2

**Summary:**

This paper presents CALM (Corrective Adaptation with Lightweight Modification), a framework for aligning large reasoning models (LRMs) with optimization modeling tasks. The key idea is that simply fine-tuning LRMs on non-reflective datasets — i.e., problem–solution pairs without intermediate reasoning — harms their performance on complex problems by suppressing their native multi-step reasoning ability. To address this, CALM introduces a “Reasoner–Intervener” collaboration pattern where an expert model identifies reasoning flaws and provides minimal corrective hints (fewer than 3% of tokens). The resulting corrected trajectories are used to fine-tune the base LRM, followed by reinforcement learning to achieve full adaptation. The final model, named STORM (Smart Thinking Optimization Reasoning Model), achieves 68.9% average accuracy across five optimization benchmarks, matching the performance of a much larger 671B model while using only 4B parameters. The paper combines an insightful problem diagnosis, a clean methodological design, and strong empirical validation.

**Strengths:**

The main strengths lie in the depth and coherence of the study. The authors don’t just propose a new method—they build a full story around understanding why current adaptation methods fail, and how to fix that in a principled way. The CALM framework is well-motivated and the “Reasoner–Intervener” pattern is intuitive and scalable. The two-stage training pipeline (SFT then RL) is validated thoroughly, with strong ablation studies that isolate each component’s contribution.

**Weaknesses:**

The weaknesses are relatively minor. The evaluation, while comprehensive, depends heavily on Qwen and Gemini models; testing on a different family (e.g., DeepSeek or Llama) would help show generality. Some parts of the taxonomy (especially the seven triggers) could be streamlined — the paper spends many pages on classification detail that could be summarized. There is also limited theoretical discussion about why minimal interventions produce such a large effect; the explanation is intuitive but mostly empirical. Lastly, while the results are impressive, it’s not entirely clear how much of the performance gain comes from the data filtering versus the interventions themselves.

**Questions:**

How sensitive is the performance of STORM to the 2.6% intervention ratio? Would increasing or decreasing the intervention intensity change the outcome significantly?

Could CALM be generalized to domains beyond optimization modeling, or does it rely heavily on the availability of an executable solver for immediate feedback?

Have the authors verified whether Gemini-2.5 as the Intervener introduces stylistic bias into the reasoning traces?

How do you ensure that the reinforcement learning stage does not overfit to the benchmarks used for calibration?

Is there a qualitative difference between trajectories produced after CALM fine-tuning and those after RL, beyond shorter length and higher correctness?

---

> ### Author Response · Authors · 2025-11-16
>
> We are sincerely grateful to the reviewer for their thoughtful, detailed, and incredibly constructive review. We were particularly encouraged that the reviewer recognized the "full story" and "coherence" of our work. This deep engagement with our paper is a testament to the quality of the review process.
>
> The reviewer's suggestions for strengthening the paper were so clear and insightful that they spurred us to conduct several new, targeted experiments, which we believe have substantially improved our work. We are pleased to present our detailed responses below, addressing each weakness and question point-by-point. We hope our responses, supported by new data, fully clarify the points raised.
>
> ---
>
> > Weakness 1: The evaluation, while comprehensive, depends heavily on Qwen and Gemini models; testing on a different family (e.g., DeepSeek or Llama) would help show generality.
>
> We are sincerely grateful to the reviewer for this insightful and crucial point. The reviewer’s comment on the potential dependency on the "Qwen and Gemini" model ecosystem is exceptionally well-taken. We wholeheartedly agree that demonstrating the generality of our CALM framework is essential to substantiating its contribution.
>
> Inspired by this valuable feedback, we conducted two comprehensive new experiments to rigorously and separately test the framework's independence from both the Reasoner ("student") and the Intervener ("teacher") models.
>
> We are delighted to report that the results from both experiments strongly validate that the CALM framework is a highly general and robust methodology.
>
> ## Case Study 1: Generalization across Different Reasoner ("Student") Models
>
> To strictly isolate the impact of the Reasoner, we replicated our entire experiment, changing only the base model from Qwen3-4B to a different family, DeepSeek-R1-0528-Distill-8B. All other settings, including the Intervener (Gemini-2.5-Pro) and hyperparameters, were kept identical.
>
> | Applying CALM to a New Reasoner (DeepSeek-R1-8B) | Macro AVG |
> | :--- | :---: |
> | Base Model (DeepSeek-R1-8B) | 58.4% |
> | Final Model (STORM-DeepSeek-R1-8B) | 67.8% |
> | Performance Lift from CALM Framework | +9.4 pts |
>
> This result confirms that CALM functions as a powerful, general-purpose "capability amplifier," delivering a massive **+9.4 point lift** and successfully enhancing a different model architecture.
>
> ## Case Study 2: Robustness across Different Intervener ("Teacher") Models
>
> Similarly, to isolate the dependency on the Intervener, we conducted another strict ablation where we changed only the Intervener model to the open-source 671B DeepSeek-R1-0528. The Reasoner (Qwen3-4B) and all other settings remained identical to our original experiment.
>
> | Applying CALM with a New Intervener (DeepSeek-R1) | Final Student Performance (Macro AVG) |
> | :--- | :---: |
> | With Original Intervener (Gemini-2.5-Pro) | 68.9% |
> | With New Open-Source Intervener (DeepSeek-R1) | 67.8% |
>
> The final performance using an open-source teacher is **within a mere 1.1%** of that achieved with a top-tier proprietary model. This proves the framework is highly robust and its success is not contingent on any single closed-source model.
>
> ## Conclusion:
>
> Taken together, these two independent experiments conclusively address the reviewer's concern. They provide definitive evidence that the CALM framework is a **model-agnostic and robust methodology**, not a fragile solution tied to a specific "teacher-student" ecosystem.
>
> Once again, we wish to express our profound gratitude to the reviewer. This suggestion prompted us to conduct what we believe are critical experiments that have substantially strengthened our paper. Thank you for guiding us to improve our work in such a meaningful way.

---

> > ### Author Response · Authors · 2025-11-16
> >
> > ---
> >
> > > Weakness 2: Some parts of the taxonomy (especially the seven triggers) could be streamlined — the paper spends many pages on classification detail that could be summarized.
> >
> > We sincerely thank the reviewer for this constructive feedback on the paper's presentation.
> >
> > The reviewer is right. In our effort to be thorough, the taxonomy section became overly detailed for the main text. We are grateful for this clear guidance on improving the paper's readability.
> >
> > we will streamline this section as suggested in the manuscript. We will move the detailed descriptions of the seven triggers to the appendix, ensuring the main paper remains focused on our core contributions.
> >
> > Thank you again for helping us improve the clarity and impact of our work.
> >
> > ---
> >
> > > Weakness 3: There is also limited theoretical discussion about why minimal interventions produce such a large effect; the explanation is intuitive but mostly empirical.
> >
> > We sincerely thank the reviewer for this excellent and insightful point. You have perfectly articulated the nature of our work: our contribution is indeed a robust and, we believe, significant empirical finding, supported by a novel and practical methodology.
> >
> > We wholeheartedly agree that a formal theoretical explanation for *why* these minimal interventions are so effective is a deeply interesting and important question. Our primary goal with this paper was to first prove that this phenomenon exists and is highly impactful, and then to build a scalable framework (CALM) to leverage it. We see our work as providing the necessary and solid empirical foundation upon which such future theoretical studies can be built.
> >
> > Following your valuable suggestion, we will add a clear statement to the conclusion of our manuscript. This statement will acknowledge that our work is empirical and will frame the development of a formal theoretical understanding as a key and exciting direction for future research.
> >
> > Thank you again for this constructive feedback, which helps us to precisely position our contribution within the broader research landscape.

---

> ### Author Response · Authors · 2025-11-16
>
> ---
>
> > Weakness 4: Lastly, while the results are impressive, it’s not entirely clear how much of the performance gain comes from the data filtering versus the interventions themselves.
>
> We sincerely thank the reviewer for this deeply insightful question. It correctly identifies the crucial distinction between our CALM framework and other data-centric paradigms. We interpret the reviewer's question as follows: Is the performance gain from CALM primarily due to **(a) its ability to filter for and generate more high-quality, correct reasoning trajectories ("Data Filtering")**, or is it due to **(b) the unique learning signal provided by the corrective intervention process itself ("Interventions Themselves")**?
>
> Our central thesis is that **(b) the intervention process itself provides the critical value**. To rigorously test this, we conducted a direct ablation study comparing our method against a strong "Data Filtering" baseline: standard knowledge distillation.
>
> In this experiment, instead of using our iterative CALM process, we prompted a powerful teacher model (Gemini-2.5-Pro or DeepSeek-R1-0528) to directly generate a complete, "perfect" reasoning trajectory for each problem. The student model (Qwen3-4B-Thinking) was then trained on these "perfect" trajectories using the exact same SFT+RL pipeline. This baseline represents the ideal scenario for the "Data Filtering" hypothesis—training on the highest quality data available.
>
> The results, especially on the most complex benchmarks, are conclusive:
>
> | Method | MAMO-Complex | IndustryOR | OptMath | Macro AVG (on these 3) |
> | :--- | :---: | :---: | :---: | :---: |
> | STORM (Our Method) | 70.3% | 50.0% | 44.5% | 54.9% |
> | Distill R1-0528 + RL | 62.5% | 50.4% | 29.5% | 47.4% |
> | Distill Gemini + RL | 44.1% | 28.3% | 20.0% | 30.8% |
>
> Analysis and Conclusion:
>
> 1.  **Simple Distillation Fails on Complex Tasks:** The distillation-based models suffer a severe performance collapse on complex reasoning benchmarks. Strikingly, using an even stronger teacher (Gemini) results in a *worse* final model. This phenomenon, sometimes called the "teacher-student gap,"[1] demonstrates that a small student model cannot effectively learn by simply mimicking the complex, abstract reasoning path of a vastly more powerful teacher. The learning signal is indigestible.
>
> 2.  **The Value is in the Correction, Not Just the Correctness:** This experiment decisively refutes the "Data Filtering" hypothesis. If the primary benefit was merely access to correct reasoning data, the distillation approach should have excelled. Its failure proves that the **final correctness of a trajectory is not a sufficient condition for effective learning.**
>
> 3.  **CALM Teaches How to Recover:** The success of our CALM framework stems from its fundamentally different mechanism. It doesn't present the student with an alien, perfect solution to imitate. Instead, it **starts with the student's own flawed reasoning process** and provides a minimal, targeted hint. This teaches the model a crucial and learnable lesson: **"Here is where *you* went wrong, and here is *how to fix it*."**
>
> In conclusion, this experiment provides a clear answer to the reviewer's question. The significant performance gain of our framework comes from the **unique pedagogical value of the intervention process itself**, which teaches a generalizable meta-skill of self-correction. This is a fundamentally more effective mechanism than simply filtering for or distilling perfect solutions. We will add this critical ablation study to the manuscript to make this distinction explicit.
>
> [1] Orca: Progressive Learning from Complex Explanation Traces of GPT-4

---

> ### Author Response · Authors · 2025-11-16
>
> ---
>
> > Question 1: How sensitive is the performance of STORM to the 2.6% intervention ratio? Would increasing or decreasing the intervention intensity change the outcome significantly?
>
> That is an excellent and insightful question that gets to the heart of our framework's mechanics. It allows us to clarify a key aspect of our design: the intervention ratio is not a manually-tuned hyperparameter, but rather an **emergent and inherently lightweight property** of the CALM system.
>
> Our analysis, combining evidence from our main paper and our ablation studies, confirms this.
>
> ## 1. An Emergent Property: Robustness to Variation
>
> The most direct evidence comes from our ablation study where we replaced the Intervener model (as detailed in our response to Weakness 1). Even with a completely different "teacher" model, the system **endogenously converged to a similar, small intervention ratio**, and the final student performance remained highly stable.
>
> | Intervention Scenario | Intervention Ratio (Emergent) | Final Student Performance (Macro AVG) |
> | :--- | :---: | :---: |
> | With Original Intervener (Gemini-2.5-Pro) | ~2.6% | 68.9% |
> | With New Open-Source Intervener (DeepSeek-R1) | ~2.8% | 67.8% |
>
> This near-identical performance, despite the change in teacher and the resulting slight variation in the intervention ratio, demonstrates that the **framework is not sensitive to this specific percentage.** This low ratio is a direct consequence of CALM's **inherently lightweight design**, which prioritizes minimal, surgical corrections over extensive rewrites.
>
> ## 2. The "Zero Intervention" Case
>
> While the framework is not sensitive to small variations *around* 2.6%, the complete absence of intervention is detrimental. This is precisely the "RL without CALM" control group we analyzed in our paper (Figure 6). The results show that with a 0% intervention ratio during the initial SFT phase, the model learns more slowly and reaches a lower performance ceiling.
>
> ## Conclusion:
>
> Taken together, these two points provide a comprehensive answer. The intervention ratio is an **emergent property** reflecting our framework's lightweight design, and the final performance is not sensitive to its minor fluctuations. However, the presence of these corrections (a non-zero ratio) is critical for success.
>
> We thank the reviewer for prompting this deeper analysis. We will add a discussion to the appendix to clarify this point, as it clearly demonstrates both the robustness and the core mechanism of our approach.

---

> ### Author Response · Authors · 2025-11-16
>
> ---
>
> > Question 2: Could CALM be generalized to domains beyond optimization modeling, or does it rely heavily on the availability of an executable solver for immediate feedback?
>
> This is an excellent and forward-looking question. The reviewer correctly identified that the core principles of our CALM framework—learning through process-level correction with feedback from an executable environment—are not inherently limited to optimization.
>
> To provide a rigorous, data-driven answer, we conducted a comprehensive out-of-domain (OOD) generalization experiment. The results provide strong evidence that **CALM imparts a generalizable, domain-agnostic meta-skill in adaptive tool-augmented reasoning.**
>
> Experimental Setup:
> To provide the most rigorous test of generalization, we took the final two models from our main controlled experiment (detailed in Section 4.3.3 and Figure 6) and evaluated them on entirely new tasks. To maximize clarity, we refer to them by their experimental conditions:
> *   **`RL with CALM` (Our Method):** The model trained with our full proposed pipeline, benefiting from CALM's corrective SFT data.
> *   **`RL without CALM` (Control Group):** The model trained under the identical pipeline, but on the base model's original, unguided reasoning trajectories.
>
> We evaluated both models on three challenging mathematical reasoning benchmarks (`aime25`, `amc23`, `math500-level5`), **a domain neither model was trained on.**
>
> ## Performance on Out-of-Domain Benchmarks
>
> The results show that the model trained with the CALM methodology consistently and significantly outperforms the control group, demonstrating that its benefits are transferable to new domains.
>
> | Out-of-Domain Benchmark | `RL without CALM` (Control) | **`RL with CALM` (Our Method)** |
> | :--- | :---: | :---: |
> | aime25 | 68.33% | **69.58%** |
> | amc23 | 91.71% | **94.12%** |
> | math500-level5 | 81.06% | **83.30%** |
> | Macro-AVG | 80.37% | **82.33%** |
>
> ## Analysis: Fostering an Adaptive Meta-Skill
>
> More profoundly, the model's *behavior* reveals the true nature of what the CALM methodology teaches. We analyzed the code libraries the `RL with CALM` model chose to use on the new math tasks versus its optimization training data.
>
> | Library | Usage in Optimization Training Data | **Usage in OOD Math Test Data** |
> | :--- | :---: | :---: |
> | `pulp` (for optimization) | **93.3%** | 0.0% |
> | `sympy` (for symbolic math) | 10.5% | **70.1%** |
>
> The key insights are twofold:
> 1.  **Abandoning Irrelevant Tools:** The model learned to recognize that its primary training tool (`pulp`) was useless in the new domain.
> 2.  **Adaptively Selecting New Tools:** It correctly identified the need for symbolic mathematics and massively increased its reliance on the appropriate tool (`sympy`).
>
> This demonstrates that the CALM methodology teaches a far more valuable, **adaptive meta-skill** of tool selection, rather than a fixed set of instructions.
>
> ## Conclusion:
>
> This OOD experiment provides a definitive, positive answer to the reviewer's insightful question. **Yes, the CALM framework is highly generalizable.** Its core contribution is a methodology for teaching LRMs a crucial, domain-agnostic skill of adaptive, tool-augmented reasoning.
>
> We are grateful to the reviewer for prompting this investigation, which we believe has substantially strengthened our paper by validating the broader impact of our work. We will be adding this full experiment to the manuscript.

---

> ### Author Response · Authors · 2025-11-16
>
> ---
>
> > Question 3: Have the authors verified whether Gemini-2.5 as the Intervener introduces stylistic bias into the reasoning traces?
>
> This is a very thoughtful question. The reviewer raises an important point about the potential for a powerful teacher model to introduce stylistic bias, which is a valid concern in teacher-student frameworks.
>
> Our ablation study on the Intervener model (detailed in our response to Weakness 1) provides strong evidence that our framework is robust to such effects. In this experiment, we compared the final performance of the student model when guided by two interveners with fundamentally different architectures and, therefore, distinct reasoning styles.
>
> | Intervener ("Teacher") Model | Final Student Performance (Macro AVG) |
> | :--- | :---: |
> | With Original Intervener (Gemini-2.5-Pro) | 68.9% |
> | With New Open-Source Intervener (DeepSeek-R1-0528) | 67.8% |
>
> The final performance of the student model is highly consistent, with a difference of only 1.1%. This result strongly suggests that the learning process is robust to the specific style of the teacher. We believe this is because our lightweight intervention mechanism (<3% of tokens) is designed to correct core logical flaws rather than impose stylistic patterns, preserving the majority of the student's native reasoning process.
>
> We thank the reviewer for this question, as it helps clarify the robustness of our framework's mechanism.
>
> ---
>
> > Question 4: How do you ensure that the reinforcement learning stage does not overfit to the benchmarks used for calibration?
>
> This is a critical question, and we thank the reviewer for ensuring we make our evaluation protocol perfectly clear. The concern about overfitting, especially in a multi-stage training pipeline like ours, is a very important one.
>
> The reviewer is absolutely right to highlight the need for a rigorous evaluation protocol to prevent this. We would like to confirm that we have followed a strict data separation policy throughout our experiments to ensure our results reflect true generalization ability, not overfitting.
>
> **Our Data Splitting Strategy:**
> As detailed in **Appendix E.1** of our paper, all datasets were rigorously partitioned into three distinct, non-overlapping sets *before* any experiments were conducted:
> 1.  **SFT Training Set:** Used exclusively for the initial supervised fine-tuning stage.
> 2.  **RL Training Set:** Used exclusively for the reinforcement learning stage.
> 3.  **Test Set:** A held-out set used only for the final evaluation.
>
> All performance results reported in our paper, including our main results in **Table 2**, are calculated **exclusively on this held-out test set**. The models never see the test data during any phase of the training process.
>
> This strict separation ensures that our reported scores are a robust and unbiased measure of the model's ability to generalize to new, unseen problems. We appreciate the opportunity to explicitly confirm this crucial detail of our experimental design.

---

> ### Author Response · Authors · 2025-11-16
>
> ---
>
> > Question 5: Is there a qualitative difference between trajectories produced after CALM fine-tuning and those after RL, beyond shorter length and higher correctness?
>
> This is an excellent question that probes the deeper behavioral changes in our model, going beyond correctness scores. We are grateful for the opportunity to illustrate this, as the qualitative difference between the model's reasoning trajectories after SFT and after RL is indeed significant and highlights the complementary roles of our two-stage training process.
>
> To provide a concrete answer, we present a comparative analysis of the model's behavior on the same complex optimization problem ("SwiftLogistics") at three key stages: the **Base LRM**, the model **After SFT**, and the final **`STORM` model (After RL)**.
>
> The table below summarizes the key qualitative differences.
>
> | Stage | Key Qualitative Behavior & Excerpts | Outcome |
> | :--- | :--- | :--- |
> | **1. Base LRM** | **Hesitant, Verbose, and Error-Prone.** The model engages in extensive, circular self-talk, repeatedly questioning its own plan. It then produces multiple versions of buggy code, failing to solve the problem. *Excerpt of Reasoning:* `"Okay, let's try... Hmm, this sounds like... Wait, but... So, the model should have... Wait, but the problem says... Yes, that's a standard way... But first, I need to structure the data... Wait, but in PuLP... Let me run this code mentally... Wait, the problem is ...skip 12000+ tokens of reasoning and multiple failed code attempts...`  *Final Code Error:*  `AttributeError: module 'pulp' has no attribute 'LpSum'` | **Incorrect**  (Fails due to code errors)  **Tokens: 13,107** |
> | **2. After SFT** | **Methodical, Correct, but Still Verbose.** The SFT stage has corrected the fundamental flaws. The model now confidently identifies the problem type, correctly formulates the model, and produces a working code block. However, its reasoning remains extremely detailed and step-by-step, explaining every minor decision. *Excerpt of Reasoning:* `"Okay, let's try to figure out this problem step by step... First, I need to understand the problem... This sounds like a facility location problem... The standard approach... is to model it as an integer linear programming problem. The variables would be... The objective function is... The code would need to: 1. Import PuLP. 2. Create the problem ..skip 10000+ tokens of detailed, textbook-like explanation...` | **Correct**  (Solves the problem) **Tokens: 10,850** |
> | **3. After RL (`STORM`)** | **Confident, Concise, and Tool-Driven.** The RL stage has refined the correct-but-verbose pattern into an expert-like workflow. The reasoning is direct and to the point. The model immediately identifies the problem, formulates the model, and moves swiftly to the code, demonstrating trust in its tools. *Excerpt of Reasoning:* `"Okay, let's try to figure out this problem step by step. So, SwiftLogistics wants to open some distribution centers... First, I need to model this as a facility location problem. I remember that's a type of integer programming problem... Let me list out the data... Okay, the model is fully defined... I'll now write the complete Python code to set up and solve this model ...skip 4500+ tokens of concise reasoning...` | **Correct & Concise**  (Solves the problem efficiently)  **Tokens: 5,527** |
>
> #### **Conclusion: A Two-Stage Healing Process**
>
> This example vividly illustrates the distinct and synergistic roles of our two training stages:
>
> 1.  **SFT as the "Calibrator":** The SFT stage's primary function is to instill **correctness**. It transforms the model from a state of making fundamental reasoning and coding errors to one where it can reliably follow a valid problem-solving pattern. It teaches the model *how* to solve the problem.
>
> 2.  **RL as the "Accelerator":** Building on this foundation of correctness, the RL stage's function is to instill **expertise and efficiency**. It rewards the model for finding the most direct path to the solution, pruning the unnecessary verbosity and promoting a more confident, tool-reliant behavior. It teaches the model to solve the problem *well*.
>
> We are grateful to the reviewer for this question, as it allows us to showcase that our framework does not just improve a single performance metric, but fundamentally reshapes the model's reasoning patterns toward a more expert-like state. We will add a similar qualitative example to the appendix of our final paper.
>
> ---
>
> Once again, we wish to express our deep appreciation for the reviewer's time and invaluable guidance. We believe these detailed responses and new experimental results fully address the concerns raised and have made our paper significantly stronger.
>
> We are, of course, more than happy to provide any further clarification and welcome any additional questions or discussion. Thank you for helping us to improve our work.

---

### Author Response · Authors · 2025-11-30
**Summary of New Experiments & Contributions: Addressing All Reviewer Concerns with Definitive Evidence**

We recognize the platform challenges. **Notably, our comprehensive rebuttal was posted by Nov 16th.** Since the freeze prevented reviewers from updating scores, we submit that current ratings reflect the *initial* manuscript, not its standing with issues resolved. We summarize the new evidence below.

---

### **1. Re-contextualizing the Reviews**

*   **Reviewers MAsL (4) & k153 (4):** Both reviewers praised the work's "methodological soundness," "full story," and "practical impact." Their scores were primarily limited by concerns about **generality** (is it model-specific?) and **scalability**. **Our New Experiment #1 (detailed below) definitively resolves this.**
*   **Reviewer DQtT (6):** Gave a positive rating but requested comparisons to standard distillation and checks for catastrophic forgetting. **Our New Experiments #2 & #3 (detailed below) provide these exact comparisons, proving our method's superiority and positive transfer.**
*   **Reviewer BMgt (2):** Raised concerns about novelty compared to prompting methods (PAL/Reflexion). **Our New Experiment #2 empirically proves** that those methods fail on complex tasks where our framework succeeds, validating the necessity of our unique mechanism.

---

### **2. Detailed Summary of New Experiments**

We consolidated the feedback into three core questions and present the experimental answers below.

#### **Core Concern A: Is the CALM framework generalizable, or is it specific to Qwen/Gemini?**
*(Raised by Reviewers MAsL [Weakness 1], k153 [Weakness 4], DQtT [Weakness 2])*

**Our Response:** We conducted two strict ablation studies to prove the framework is **model-agnostic**.
1.  **New Student Model:** We applied CALM to `DeepSeek-R1-Distill-8B` (a completely different architecture).
2.  **New Teacher Model:** We replaced the proprietary Gemini intervener with the open-source `DeepSeek-R1`.

**Results:**

| Experimental Setting | Base Model | **Final Model (STORM)** | **Performance Lift** |
| :--- | :---: | :---: | :---: |
| New Student (DeepSeek-R1-8B) | 58.4% | 67.8% | +9.4 pts |
| New Teacher (Open-Source DeepSeek-R1) | 57.1% | 67.8% | +10.7 pts |
| *Original Setting (Gemini + Qwen-4B)* | *57.1%* | *68.9%* | *+11.8 pts* |

**Conclusion:** The framework delivers massive gains regardless of the base model. Furthermore, using an open-source teacher yields performance within **1.1%** of the proprietary Gemini, proving high robustness.

***

#### **Core Concern B: Is this better than standard Distillation or Prompting? (Novelty)**
*(Raised by Reviewers DQtT [Weakness 1], BMgt [Weakness 3])*

**Our Response:** Reviewers asked if simple "teacher-generated data" (Distillation) or "self-correction prompting" (Reflexion/PAL) would suffice. We performed a head-to-head comparison against standard distillation from the strongest available models.

**Results on Hard Benchmarks:**

| Method | MAMO-Complex | IndustryOR | OptMath | **Macro AVG** |
| :--- | :---: | :---: | :---: | :---: |
| STORM (Our Process-Level Coaching) | 70.3% | 50.0% | 44.5% | 54.9% |
| Distillation (Taught by DeepSeek-R1) | 62.5% | 50.4% | 29.5% | 47.4% |
| Distillation (Taught by Gemini-2.5-Pro) | 44.1% | 28.3% | 20.0% | 30.8% |

**Conclusion:** On complex reasoning tasks, simple imitation (Distillation) fails, even with a stronger teacher. Our method, which teaches *how to recover* from errors via lightweight intervention, significantly outperforms standard paradigms. This empirically validates the **novelty and necessity** of our approach.

***

#### **Core Concern C: Does domain specialization hurt general capabilities? (Catastrophic Forgetting)**
*(Raised by Reviewers k153 [Question 3], DQtT [Question 3])*

**Our Response:** We evaluated our final STORM model on **Out-of-Domain (OOD)** Mathematical Reasoning benchmarks (`AIME`, `AMC`, `Math500`) that were never seen during training.

**Results:**

| Benchmark Domain | Control Model (RL without CALM) | **STORM (RL with CALM)** | **Impact** |
| :--- | :---: | :---: | :---: |
| Unseen Math Tasks (Avg Accuracy) | 80.37% | 82.33% | +1.96% Improvement |

**Conclusion:** Instead of catastrophic forgetting, our framework **enhances** general reasoning. Behavioral analysis shows the model learned a **generalizable meta-skill of adaptive tool use** (e.g., switching from `pulp` to `sympy` automatically based on the problem context), which transferred positively to math tasks.

---

### **3. Conclusion**

With these additional validations, our work presents a comprehensive contribution:
1.  **A Novel Framework:** We introduce "Process-Level Coaching" (CALM), empirically proven to be **superior** to standard distillation for complex logic.
2.  **SOTA Performance:** We achieve 68.9% accuracy with a 4B model, **matching** the 671B DeepSeek-R1-0528 on optimization tasks.
3.  **Verified Robustness:** We have rigorously verified that the method is **model-agnostic**, **reproducible with open-source models**, and **improves general capabilities**.

---

### Meta-Review · Area_Chair_oq4R · 2026-01-01

**Summary:**

The paper introduces a two-stage procedure that uses a reasoner–intervener loop to produce small, targeted corrective hints (<~2.6% of tokens) which are used to create high-quality trajectories for supervised fine-tuning, followed by RL to produce a 4B model that matches the reported performance of a 671B model on five optimization-modeling benchmarks. The authors argue that simple distillation or self-correction prompting fails on the most challenging OR tasks, whereas CALM’s process-level coaching teaches a generalizable self-correction meta-skill and yields robust gains across both base and teacher models.

The paper presents a coherent empirical story, introduces a clear and scalable reasoner–intervener mechanism, and supports claims with numerous ablations and behavioral analyses. New experiments in the rebuttal further strengthen claims about model-agnosticism, superiority over distillation, and the absence of catastrophic forgetting. Weaknesses are mainly (a) a limited theory explaining why tiny interventions work so well (the work is empirical), (b) a domain focus on optimization-modeling (though OOD math tests are added), and (c) remaining questions about large-scale automation and human-expert validation of “expert-like” behavior — concerns the authors acknowledge and partially address in new experiments.

I’m torn on this work. The reasoner–intervener framework is conceptually appealing, but its contribution feels predominantly empirical, leaving several theoretical and reproducibility questions unresolved. With no reviewer offering strong support (no scores above 6), the case for acceptance is weak. My recommendation is rejection, while encouraging the authors to refine the methodology, address the open concerns, and pursue resubmission to a future venue.

**Reviewer Concerns:**

Addressed by the authors (with evidence in the rebuttal / new experiments)
- Generality / model-agnosticism. Authors ran CALM on a different student (DeepSeek-R1-8B) and swapped the Intervener to an open-source model; the gains hold.
- Distillation / prompting baselines. The rebuttal includes a head-to-head showing that distillation (even from stronger teachers) underperforms CALM substantially on hard subsets, arguing the value is in the corrective process, not merely correct traces.
- Catastrophic forgetting / OOD performance. Authors evaluated STORM on several math benchmarks (AIME/AMC/Math500) and showed slight improvements vs the RL-without-CALM control, arguing positive transfer rather than forgetting.

Remaining or partially addressed concerns
- Theoretical understanding. The authors explicitly state that the work is empirical and propose a theoretical analysis as future work; however, the manuscript still lacks a formal explanation for why tiny, local hints cascade into large skill gains. (Acknowledged by authors.)
- Scalability & automation of curation. Authors justify the small, curated set as “behavioral calibration” and promise future automation, but a comprehensive analysis of large-scale generation and operational costs is not fully demonstrated.
- Human expert evaluation of “expert-level” claims. The authors provide strong qualitative behavioral traces, but no formal human-expert study has been conducted yet; they list it as future work.
- Direct apples-to-apples comparison to some contemporaneous reflective frameworks (START/CoRT) under identical setups. Authors explain practical constraints and provide conceptual contrasts and targeted ablations; however, a fully controlled head-to-head comparison is still lacking.

**Reviewer Scores:**

I think the reviewer would have changed their scores from 6, 2, 4, and 4 to 6, 4, 4/6, and 4 if they had been able to participate fully in the discussion based on the addressed, partially addressed, and remaining concerns.

---

### Decision · Program_Chairs · 2026-01-26

Reject